# Systematic Reward Gap Optimization for Mitigating VLM Hallucinations

**Lehan He**[1,2*], **Zeren Chen**[1,3*], **Zhelun Shi**[1], **Tianyu Yu**[4], **Jing Shao**[2,3†], **Lu Sheng**[1†]

[1]School of Software, Beihang University    [2]Shanghai Innovation Institute
[3]Shanghai AI Laboratory    [4]Tsinghua University

helehan,czr1604,lsheng@buaa.edu.cn, shaojing@pjlab.org.cn

Codes and models: https://tpr-dpo.github.io

## Abstract

The success of Direct Preference Optimization (DPO) in mitigating hallucinations in Vision Language Models (VLMs) critically hinges on the true reward gaps within preference pairs. However, current methods, typically relying on ranking or rewriting strategies, often struggle to optimize these reward gaps in a systematic way during data curation. A core difficulty lies in precisely characterizing and strategically manipulating the overall reward gap configuration, that is, the deliberate design of how to shape these reward gaps within each preference pair across the data. To address this, we introduce Topic-level Preference Rewriting (TPR), a novel framework designed for the systematic optimization of reward gap configuration. Through selectively replacing semantic topics within VLM responses with model's own resampled candidates for targeted rewriting, TPR can provide topic-level control over fine-grained semantic details. This precise control enables advanced data curation strategies, such as progressively adjusting the difficulty of rejected responses, thereby sculpting an effective reward gap configuration that guides the model to overcome challenging hallucinations. Comprehensive experiments demonstrate TPR achieves state-of-the-art performance on multiple hallucination benchmarks, outperforming previous methods by an average of $\sim$20%. Notably, it significantly reduces hallucinations by up to 93% on ObjectHal-Bench, and also exhibits superior data efficiency towards robust and cost-effective VLM alignment.

## 1 Introduction

Vision language models (VLMs) [1, 2, 3, 4, 5] have achieved remarkable success across a spectrum of multimodal tasks, from visual question answering [4, 6] to image captioning [7, 8], becoming foundational components in modern AI systems. Despite these advancements, even leading models like GPT-4V [1] suffer from a critical limitation: visual hallucinations [9, 10, 11, 12]. Specifically, they might confidently describe non-existent objects, misrepresent attributes, or misjudge spatial relationships, contradicting visual inputs. This poses significant risks, especially in safety-critical scenarios such as autonomous driving [13] and medicine applications [14].

Recent efforts to mitigate visual hallucinations [12, 15, 16, 17, 18, 19] increasingly leverage preference learning through alignment techniques such as Direct Preference Optimization (DPO) [20]. These methods aim to steer VLM behaviors towards desired outcomes by learning from meticulously curated preference data. This data typically consists of preference pairs $(y_w, y_l)$ for a given input $x$, where response $y_w$ is preferred over $y_l$. By aligning with these preference data, the policy learned by DPO implicitly defines a reward function $r(y; x)$ that reflects a probabilistic model like Bradley-Terry $p(y_w \succ y_l | x)$ [21]. The quality of this learned reward function, as highlighted by

---

*Equal contribution.    †Corresponding author.

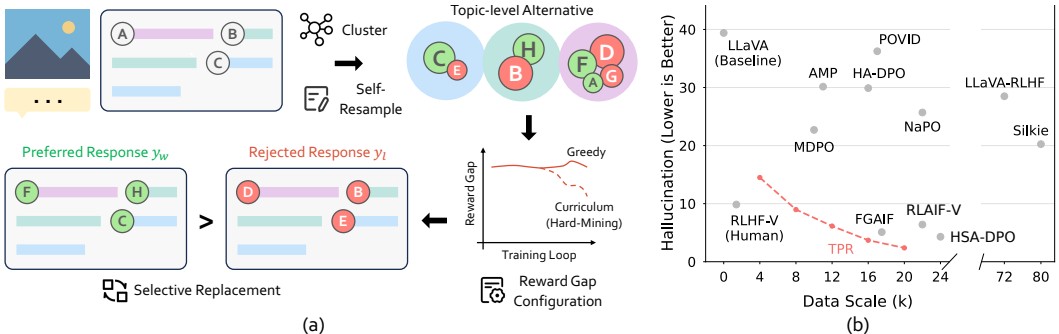

Figure 1: **(a) Topic-level Preference Rewriting.** Based on varying chosen strategies, TPR selectively replaces each topic using model's internally resampled candidates. Here, "Greedy" denotes selecting the highest- and worst-scored alternatives for a high-divergence reward gap, while "Curriculum" gradually introduces harder-to-discern hallucinations in $y_l$, thereby adjusting the reward gap to master challenging and subtle hallucinations. **(b) Data Efficiency.** Apart from manual annotation (RLHF-V [12]), TPR achieves the best data efficiency on visual hallucination reduction.

recent studies [19, 22, 23], hinges on the fidelity and magnitude of true reward gaps instantiated by each preference pair in the curated data. Thus, during data curation, strategically designing the reward gaps within each preference pair to shape an effective learning trajectory for the target reward functions, is crucial for robust VLM alignment against hallucinations. We refer to this deliberate process as the systematic optimization of overall ***reward gap configuration***. Given that these reward gaps are inherently shaped by the intrinsic data characteristics of $y_w$ and $y_l$ (such as informativeness, trustworthiness, and explicit differences), optimizing reward gap configuration necessitates more than mere preference collections, but carefully controlling the underlying data characteristics, thereby sculpting an optimal reward function tailored for minimizing hallucinatory outputs.

However, existing methods for preference data curation often lack mechanisms for such deliberate reward gap optimization. For example, ranking-based methods [15, 17, 19, 24] directly select $y_w$ and $y_l$ from potentially flawed model outputs without correcting underlying hallucinations. This may lead to low informativeness or insufficient reward gaps to penalize hallucinations, ultimately providing weak signals for learning an effective reward function. Alternatively, rewriting-based approaches, particularly those [58, 26, 27] employing external "black-box" models like GPT-4V [1], encounter difficulties in precisely adjusting the generated responses (*e.g.*, the type and magnitude of the changes) and risk introducing hallucinations in $y_l$ that deviate from model's intrinsic failure modes. Consequently, both approaches may yield a suboptimal reward gap configuration across the curated data, compromising the learning of robust reward functions.

To address these limitations, we propose **T**opic-level **P**reference **R**ewriting (**TPR**), a novel VLM hallucination mitigation paradigm designed for systematically optimizing reward gap configuration during data curation, as illustrated in Figure 1 (a). To provide precise, fine-grained control over semantic details between $y_w$ and $y_l$, TPR operates at the topic-level. Specifically, TPR first decomposes responses into semantic units and clusters them into distinct topics. It then performs intra-topic self-resampling using the model itself to avoid introducing external biases that may occur with rewriting-based methods. Preference pairs $(y_w, y_l)$ are subsequently constructed by selectively replacing original semantic units with these alternatives from the same topic. This mechanism allows for precise, fine-grained adjustment of semantic details between $y_w$ and $y_l$, guided by various chosen strategies. For example, a greedy strategy can establish highly discriminative reward gaps by constructing $y_w$ and $y_l$ using the highest- or lowest-scored alternatives from their respective topics.

Moreover, this flexible control over fine-grained differences uniquely enables the formulation and investigation of sophisticated strategies for achieving an optimal reward gap configuration. In this work, we exemplify this capability with a simple yet effective curriculum learning strategy that progressively adjusts the difficulty of hallucinations included in the $y_l$. By implementing such hard negative mining, the model can be trained to more effectively counteract subtle and challenging hallucinations, showcasing the effectiveness of TPR. Crucially, this effectiveness is coupled with superior data efficiency, stemming directly from TPR's ability to curate high-quality preference pairs. This enhanced data efficiency, clearly depicted in Figure 1 (b), underscores how TPR's systematic optimization of reward gap configuration leads to more performant and cost-effective VLM alignment.

In summary, our contributions can be summarized as below:

- We underscore the importance of systematically optimizing reward gap configuration during data curation for robust VLM alignment, an aspect often overlooked in prior studies.

- We propose Topic-level Preference Rewriting (TPR), a novel paradigm designed to offer fine-grained control over individual reward gaps during data curation, achieved through topic-level selective replacement with the model's own resampled candidates.

- Benefiting from the fine-grained control afforded by TPR, we introduce a curriculum learning strategy that optimizes the overall reward gap configuration by progressively adjusting the difficulty of rejected responses $y_l$, enhancing robustness against challenging hallucinations.

- Comprehensive experiments demonstrate that TPR achieves the state-of-the-art performance on multiple visual hallucination benchmarks, outperforming previous methods in both performance (by $\sim 20\%$) and data efficiency.

## 2    Related Work

**Vision Language Models and Hallucinations.** The advent of Large Language Models (LLMs) [28, 29, 30, 31] has driven the development of Vision Language Models (VLMs) [1, 2, 4, 5]. Through multimodal alignment followed by supervised instruction tuning, VLMs achieve remarkable proficiency in visual perception and comprehensive understanding. However, their tendency to produce hallucinations [9, 10, 11, 32], *i.e.*, generating responses that not factually grounded in the given image, undermines their reliability and practicality in real-world applications. These hallucinations can be attributed to multiple factors, including inherent biases inherited from LLMs [33], biased training data [34, 35], insufficient multimodal alignment [36] and suboptimal inference strategies [37]. Efforts to mitigate hallucinations broadly fall into training-free and training-based approaches.

**Training-free Hallucination Reduction.** Training-free approaches [38, 39, 40, 41, 42] aim to reduce hallucinations without additional model training, typically by intervening during the inference stage. These approaches often involve adjusting decoding strategies or implementing post-hoc output correction mechanisms. For example, HallE-Switch [38] modifies the decoding process to suppress object predictions with low confidence scores. MARINE [39] employs classifier-free guidance using auxiliary object grounding features to enrich the visual context during generation. Woodpecker [40] operates post-hoc, identifying and rectifying factual inconsistencies in generated text by leveraging feedback from a more capable VLM. While efficient, these methods primarily address the symptoms rather than the core deficiencies of hallucinations within the model itself and may offer limited improvements against deeply ingrained hallucinatory tendencies.

**Training-based Hallucination Reduction.** Training-based methods [12, 15, 16, 17, 18, 19, 27, 43] predominantly learning from preference through alignment techniques such as Direct Preference Optimization (DPO) [20]. Preference data curation is central to these methods, as the policy's learned behavior depends significantly on the quality of the training supervision embedded in the preference data. To construct high-quality preference data, these methods typically adopt ranking-based or rewriting-based strategies. Ranking-based methods [17, 19, 24] often employ an auxiliary labeler model to distinguish preferred responses. For example, RLAIF-V [17] implements a divide-and-conquer strategy that aggregates scores from decomposed sub-responses, reducing dependence on proprietary models. AMP [19] constructs multi-level preferences by contrasting outputs from models of varying scales, enabling cross-level comparison. On the other hand, rewriting-based methods involve modifying an initial response to create a preferred response $y_w$ or a rejected response $y_l$. This rewriting can be performed manually by human annotators [12, 16] or automated using AI rewriters [58, 26, 27] like GPT-4V [1]. Exploring effective data curation approaches remains an active area of research, motivating the development like TPR proposed in this work.

## 3    Topic-level Preference Rewriting

### 3.1    Preliminary: Preference Data and Alignment

Mitigating visual hallucinations in VLMs often involves aligning a policy model $\pi_\theta$ with carefully curated preference data, denoted as $\mathcal{D} = \{(I, x, y_w, y_l)\}$. Typically, for a given image $I$ and prompt

$x$, these methods employ a base reference model $\pi_{\text{ref}}$ to generate candidate responses. Human experts or proxy AI labelers $\pi_{\text{label}}$ then evaluate these responses or rewrite them to form preference pairs $(y_w, y_l)$, where $y_w$ is preferred over $y_l$. The target policy model $\pi_\theta$, often initialized from $\pi_{\text{ref}}$, is subsequently fine-tuned on $\mathcal{D}$ using Direct Preference Optimization (DPO) [20].

The core idea of DPO is to optimize the policy $\pi_\theta$ to satisfy the preferences in $\mathcal{D}$, which are assumed to follow a latent reward model that reflects the preference probability $p(y_w \succ y_l | x)$, while simultaneously being constrained by a KL-divergence penalty to not stray too far from the initial reference policy $\pi_{\text{ref}}$. The DPO loss function is formulated as:

$$\mathcal{L}_{\text{DPO}}(\pi_\theta; \pi_{\text{ref}}) = -\mathbb{E}_{(x,y_w,y_l)\sim\mathcal{D}} \left[ \log \sigma \left( \beta \log \frac{\pi_\theta(y_w|x)}{\pi_{\text{ref}}(y_w|x)} - \beta \log \frac{\pi_\theta(y_l|x)}{\pi_{\text{ref}}(y_l|x)} \right) \right] \quad (1)$$

where $\beta$ is a hyperparameter that controls the strength of the preference modeling versus the KL constraint.

As highlighted by recent studies [19, 22, 23], the effectiveness of such alignment critically depends on the true reward gaps, formally defined as the difference $r(y_w; x) - r(y_l; x)$ within each preference pair. The strategic configuration of reward gaps across the curated preference data provides substantial training signals that not only reinforce desired behaviors but also accurately expose the model's genuine and harder-to-discern deficiencies. Therefore, to sculpt effective reward gaps, two key principles should guide data curation: **(1)** The process must enable fine-grained control over data characteristics in $y_w$ and $y_l$ to deliberately shape reward gaps. **(2)** The preference pairs should accurately reflect desired behaviors while exposing model's intrinsic failure modes.

Guided by these principles, we propose Topic-level Preference Rewriting (TPR), a novel framework designed to systematically configure the reward gap by establishing fine-grained control over topic-level semantic details within the response. TPR implements this through two core steps: topic-level alternatives generation (Section 3.2) and selective topic replacement (Section 3.3). Building upon this precise control, TPR enables the active exploration and strategic shaping of reward gap configuration, exemplified by the curriculum learning strategy detailed in Section 3.4. For clarity, we provide a detailed pseudo code of the complete TPR workflow in Algorithm 1.

---

**Algorithm 1** Topic-level Preference Rewriting (TPR)

**Require:** Reference model $\pi_{\text{ref}}$, Labeler model $\pi_{\text{label}}$, Source data $\mathcal{D}_{\text{src}}$ (Image $I$, Prompt $x$), Chosen strategy $\omega$ (e.g., greedy, curriculum).
**Ensure:** Preference data $\mathcal{D}_{\text{pref}} = \{(I, x, y_w, y_l)\}$.
1: Initialize $\mathcal{D}_{\text{pref}} \leftarrow \varnothing$;
2: **for** each $(I, x)$ in $\mathcal{D}_{\text{src}}$ **do**
3:  Initialize initial responses $S_y \leftarrow \varnothing$, semantic units $S_u \leftarrow \varnothing$, topic clusters $S_C \leftarrow \varnothing$;
4:  **for** $i \leftarrow 1$ to $M$ **do**
5:   $y_i \leftarrow \text{Sample}(\pi_{\text{ref}}, I, x)$;
6:   Add $y_i$ to $S_y$;
7:   Add $\text{Decompose}(\pi_{\text{ref}}, y_i)$ to $S_u$;
8:  **end for**
9:  $S_C \leftarrow \text{TopicCluster}(\pi_{\text{ref}}, S_u)$;
10:  **for** each cluster $C$ in $S_C$ **do**
11:   $\text{IntraTopicResample}(\pi_{\text{ref}}, C)$;
12:   $\text{Rank}(\pi_{\text{label}}, C)$;
13:  **end for**
14:  Initialize response template $y_k \leftarrow$ Randomly select from $S_y$;
15:  Initialize replacements $S_w \leftarrow \varnothing, S_l \leftarrow \varnothing$;
16:  **for** each unit $u_k \in C$ in $y_k$ **do**
17:   $(S_w, S_l) \leftarrow \text{SelectAlternatives}(\omega, C)$;
18:  **end for**
19:  $y_w \leftarrow \text{InContextRewrite}(\pi_{\text{ref}}, y_k, S_w)$;
20:  $y_l \leftarrow \text{InContextRewrite}(\pi_{\text{ref}}, y_k, S_l)$;
21:  Add $(I, x, y_w, y_l)$ to $\mathcal{D}_{\text{pref}}$;
22: **end for**
23: **return** $\mathcal{D}_{\text{pref}}$

---

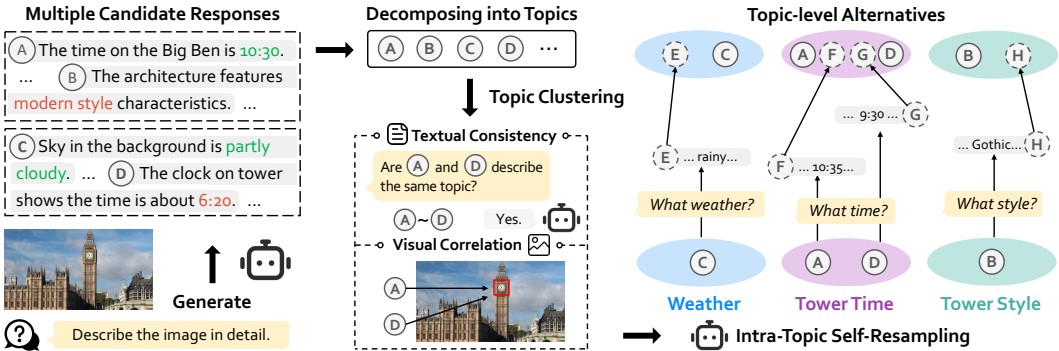

Figure 2: **Obtaining High-quality and Diverse Topic-level Alternatives.** Initially, candidate responses from VLM are decomposed into fine-grained semantic units. These units are then grouped into distinct topic clusters based on textual consistency and visual correlation. A diverse pool of topic-level alternatives is then generated via intra-topic self-resampling.

## 3.2 Topic-level Alternatives Generation

VLM responses comprise various semantic topics, encompassing diverse objects and attributes, intricate spatial relationships, or subtle contextual implications. To offer flexible control over fine-grained details within these responses, TPR operates precisely at the topic level. Such a topic-centric approach is made viable by findings like those in RLAIF-V [17], which suggest topics within a response often exhibit weak correlations, permitting their relatively independent manipulation. As illustrated in Figure 2, TPR introduces a topic-level alternative generation approach designed to provide a rich set of candidates for the subsequent selective replacement of individual semantic topics.

**Decomposing into Topics.** Following established protocol [12, 17], for an input image $I$ and prompt $x$, we begin by sampling multiple candidate responses $\{y_1, \ldots, y_M\}$ from the reference model $\pi_{\text{ref}}$. We then utilize $\pi_{\text{ref}}$ to decompose each candidate response $y_m$ into a set of fine-grained semantic units $\{u_{m,1}, \ldots, u_{m,N_m}\}$, where each semantic unit $u_{m,n}$ corresponds to a distinct semantic topic.

**Topic Clustering.** To facilitate subsequent selective replacement of semantic units with alternatives under the same topic, we group semantically related units from different candidate responses into unique topic clusters, based on textual consistency and visual correlation. Textual consistency between units $u_{m,n}$ and $u_{p,q}$ is assessed by querying the reference model $\pi_{\text{ref}}$, *e.g.*, "Are $u_{m,n}$ and $u_{p,q}$ describing the same topic?". For visual correlation, we utilize features from the VLM's visual encoder (*e.g.*, CLIP [44]) to verify whether both units refer to similar regions within the input image $I$. It is essential for disambiguating textually similar units that describe visually distinct entities. Units $u_{m,n}$ and $u_{p,q}$ are considered the same topic only if they satisfy both textual consistency and visual correlation criteria. We then apply a greedy algorithm [45] for topic clustering, yielding topic cluster $\{c\}$, each containing textually and visually related units. More details are provided in Appendix B.

**Intra-Topic Self-Resampling.** To enrich the pool of alternatives within each topic cluster, we employ the reference model $\pi_{\text{ref}}$ to perform self-resampling focused on individual topics. Specifically, for each decomposed unit $u_{m,n}$, we prompt $\pi_{\text{ref}}$ to first convert the unit into a relevant wh-question (*e.g.*, *"The time on the Big Ben is 3:30."* $\rightarrow$ *"What time is on the Big Ben?"*). We then query $\pi_{\text{ref}}$ multiple times with these topic-specific questions to obtain a set of candidate semantic units pertinent to that topic, which avoids introducing potential biases or hallucinations from external models. Compared to resampling entire responses, this intra-topic self-resampling offers two key advantages central to TPR. First, by focusing $\pi_{\text{ref}}$ on one topic at a time, it boosts the efficiency of obtaining valid and diverse semantic units, bypassing the demands of simultaneous correctness across all units required in entire response resampling. Second, generating alternatives at the topic level provides the necessary granularity for the subsequent selective replacement, enabling systematic shaping of the desired reward gaps in the resulting preference pairs.

## 3.3 Selective Topic Rewriting

**Intra-Topic Ranking.** To distinguish between accurate and potentially hallucinatory semantic units within each topic cluster $c$, we adopt an intra-topic ranking strategy. For each semantic unit

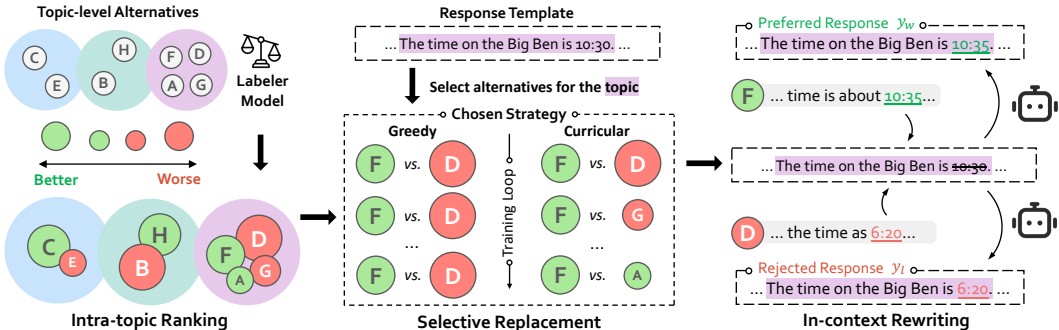

Figure 3: **Constructing Preference Pairs by Selectively Replacement.** Starting with a pool of scored topic alternatives, preference pairs are constructed through selectively replacing units within the template. The specific alternatives are chosen based on strategies like greedy replacement, to deliberately control over the resulting reward gap. In-context rewriting is then employed to seamlessly integrate these chosen alternatives, ensuring fluent preference pairs for subsequent model alignment.

$u_{m,n}^c$ (including original and self-resampled alternatives), we prompt the $\pi_{\text{ref}}$ to convert it into a corresponding yes-no question (*e.g.*, *"The time on the Big Ben is 3:30."* → *"Is the time on the Big Ben 3:30?"*). Given the input image $I$, we then query $\pi_{\text{label}}$ with converted yes-no questions, obtaining probabilities $p_Y$ and $p_N$ for "Yes" or "No" responses, respectively. Each unit $u_{m,n}^c$ is assigned a score $S(u_{m,n}^c) = p_Y - p_N$, where a higher score indicates a higher likelihood that the unit is factually accurate and non-hallucinatory for topic $c$. Moreover, as noted in prior work [17], evaluating fine-grained units tends to yield more reliable assessments, allowing even moderately capable models (*e.g.*, $\pi_{\text{ref}}$ itself) to serve effectively as $\pi_{\text{label}}$ in TPR.

**Selective Replacement.** Leveraging the ranked semantic units within each topic, we construct preference pairs $(y_w, y_l)$ through a selective replacement mechanism. First, we randomly select one of the initial candidate responses $\{y_1, y_2, \ldots, y_M\}$ as a response template, denoted $y_k$. This template consists of its original semantic units $\{u_{k,1}^{c_1}, u_{k,2}^{c_2}, \ldots\}$, each associated with a topic. We then generate $y_w$ and $y_l$ by selectively replacing units $u_{k,n}^{c_n}$ with alternative units chosen from the ranked pool within the same topic cluster $c_n$. This selective replacement is the core mechanism for providing fine-grained control over the semantic difference between $y_w$ and $y_l$. For example, a common strategy aimed at constructing highly discriminative pairs involves replacing each $u_{k,n}^{c_n}$ within the template with the highest- and lowest-scoring alternatives from their respective topic clusters. By varying chosen strategies for replacement units, this mechanism allows for deliberate adjustment of each semantic unit, consequently, the reward gap encoded within each $(y_w, y_l)$ pair.

**In-Context Rewriting.** Directly substituting alternative units into the template could potentially disrupt the natural language flow and stylistic consistency, producing awkward or incoherent responses. To mitigate this issue, we perform the selective replacement using in-context rewriting guided by the reference model $\pi_{\text{ref}}$. Specifically, we instruct $\pi_{\text{ref}}$ to integrate the semantic content of the chosen replacement units into the template response, preserving logical structure and linguistic style. This results in preference pairs $(y_w, y_l)$ where the semantic differences related to specific topics are precisely controlled while maintaining overall response quality. These high-quality preference pairs, encoding intentionally designed reward gaps, are then used to fine-tune the policy model $\pi_\theta$ via DPO, effectively steering it away from generating visual hallucinations.

## 3.4 Exploring Optimal Reward Gap Configuration

Constructing preference pairs $(y_w, y_l)$ where corresponding semantic units exhibit maximal score divergence provides a strong initial signal in DPO training. However, recent studies [19, 23] suggest such a greedy strategy may not yield the optimal reward gap configuration for robust alignment. Focusing exclusively on maximizing individual reward gaps might prioritize penalizing obvious, easily detectable errors over the subtle, challenging hallucinations that the model genuinely struggles with, potentially leading to inefficient learning. The systematic control over fine-grained semantic difference afforded by TPR allows us to explicitly optimize the reward gaps embedded in the curated data, thereby refining the model's ability to avoid these harder-to-discern failure modes.

Table 1: **Experimental Results on Several Hallucinations and General Capabilities Benchmarks.**
The best and second best results are shown in **bold** and underlined, respectively.

| | Hallucination Benchmarks | | | | | | | | | | General Benchmarks | |
| Model | ObjHal | | MMHal | | AMBER | | POPE | | RefoMB | | LLaVA-B | MMstar |
| | $CH_s\downarrow$ | $CH_i\downarrow$ | Score ↑ | Hall. ↓ | Acc. ↑ | F1 ↑ | Adv. ↑ | All ↑ | Trust. ↑ | Win. ↑ | Acc. ↑ | Acc. ↑ |
|---|---|---|---|---|---|---|---|---|---|---|---|---|
| LLaVA-RLHF-13B [15] | 38.1 | 18.9 | 2.02 | 62.5 | 79.7 | 83.9 | 82.3 | 81.9 | 26.3 | 17.2 | 61.5 | 34.2 |
| RLHF-V-13B [12] | 12.2 | 7.5 | 2.45 | 51.0 | 72.6 | 75.0 | 80.5 | 81.9 | 41.4 | 17.7 | 51.4 | 33.2 |
| Silkie-10B [24] | 27.1 | 13.4 | **3.19** | 32.3 | 82.2 | 87.6 | 80.3 | 81.1 | 38.9 | 21.2 | **73.2** | 33.6 |
| POVID-7B [26] | 48.1 | 24.4 | 2.08 | 56.2 | **82.9** | 87.4 | 84.0 | 85.8 | 44.4 | 13.6 | 62.2 | **34.3** |
| MFPO-7B [57] | 13.4 | 6.6 | 2.69 | 49.0 | – | – | – | – | – | – | – | – |
| HA-DPO-7B [58] | 39.9 | 19.9 | 1.98 | 60.4 | 75.2 | 79.9 | 82.5 | 86.9 | 39.9 | 17.2 | 67.2 | 32.9 |
| OPA-DPO-7B [59] | 13.0 | 4.3 | 2.83 | 45.8 | 81.3 | 85.6 | 83.7 | 86.1 | 39.4 | 18.2 | 62.2 | 32.2 |
| mDPO-7B [60] | 35.7 | 9.8 | 2.39 | 54.2 | – | – | – | – | – | – | – | – |
| AMP-MEG-7B [19] | 37.8 | 22.5 | 3.17 | 35.0 | 78.3 | 83.6 | 83.4 | 86.8 | 42.9 | 18.7 | 54.6 | 27.5 |
| RLAIF-V-7B [17] | 8.5 | 4.3 | 3.06 | **29.2** | 76.8 | 84.5 | 81.2 | 83.3 | 47.5 | 20.7 | 64.9 | 31.8 |
| FGAIF-7B [18] | 6.2 | 3.9 | 3.09 | 36.0 | – | – | 79.9 | 83.4 | – | – | – | – |
| HSA-DPO-13B [27] | 5.3 | 3.2 | 2.61 | 48.0 | – | – | – | – | – | – | – | – |
| LLaVA-1.5-7B [6] | 53.6 | 25.2 | 2.36 | 51.0 | 73.5 | 77.6 | **84.5** | 85.9 | 30.8 | 12.1 | 59.7 | 30.3 |
| + TPR-7B | 4.0 | 2.2 | 3.01 | 31.2 | 82.3 | 87.6 | 83.5 | 86.2 | 58.1 | 31.3 | 69.2 | 33.2 |
| + TPR-CL-7B | **3.4** | **1.8** | 3.06 | 30.2 | 82.7 | **87.8** | 84.2 | **87.6** | **61.1** | **32.3** | 71.1 | 33.3 |

Leveraging this control, we propose a simple yet effective curriculum learning strategy, analogous to hard negative mining in other domains, to optimize the reward gap configuration for hallucination reduction. This strategy involves structuring the preference data curation and DPO training in stages. Specifically, during an initial "**Warm-Up**" stage, we adopt the greedy strategy described above, constructing $(y_w, y_l)$ pairs with maximal score divergence between corresponding semantic units. This provide a strong initial learning signal, encouraging faster convergence by clearly differentiating between highly discriminative content. In a subsequent "**Hard-Mining**" stage, we gradually increase the difficulty of the learning objective by constructing $y_l$ using incorrect alternative units that have progressively higher scores, *i.e.*, they are less obviously wrong and closer to the decision boundary. Introducing these "hard negative" examples in $y_l$ challenges the model to make finer distinctions during DPO training. By learning to differentiate preferred responses from subtly flawed ones, the model can more effectively refine its ability to detect and avoid nuanced hallucinations, ultimately improving robustness. This curriculum strategy systematically shape the overall reward gap configuration over time, starting with broad distinctions and moving towards finer-grained ones ultimately. Further details are provided in Appendix C.

# 4 Experiments

## 4.1 Experimental Setup

**Models.** In line with previous studies, we use LLaVA-1.5-7B [6] as both the reference model $\pi_{\text{ref}}$ for generating preference data, and as the policy model $\pi_\theta$ that is subsequently fine-tuned. The labeler model $\pi_{\text{label}}$, used for intra-topic ranking described in Section 3.3, is LLaVA-NeXT-34B [3], a choice consistent with recent approaches [17, 27]. In our experiments, we evaluate two variants of proposed TPR: (1) **TPR** utilizes the greedy replacement strategy for data curation. (2) **TPR-CL** (**C**urriculum **L**earning) utilizes a curriculum learning strategy for data curation, detailed in Section 3.4. We compare TPR against a range of counterparts, including ranking-based methods [15, 17, 18, 19, 24] and rewriting-based methods that rely on human experts [12] or external models [26, 27, 58, 59].

**Data Source.** Following Yu, et al. [12, 17], we curate preference data based on 7 publicly available dataset sources: VQA v2 [46], MSCOCO [47], ShareGPT-4V [48], TextVQA [49], MovieNet [50], OKVQA [51] and Google Landmark v2 [52]. We generate a total of 20,000 preference data instances used for alignment. For the TPR-CL variant, 12,000 instances (60%) are constructed during "Warm-Up" stage, and the remaining 8,000 (40%) are constructed during "Hard-Mining" stage.

**Preference Learning.** We apply Direct Preference Optimization (DPO) [20] for preference learning, aligning the policy model with the preference data curated by TPR and its variants. We use the AdamW [53] optimizer with a batch size of 8, a learning rate of $5 \times 10^{-7}$ with the cosine decay strategy. The policy model is fine-tuned for 1 epoch on 8 NVIDIA A100 GPUs.

Table 2: **Ablation Studies on Different Components in TPR. Multi-Rsp.**: Sampling multiple candidate responses before decomposition. **Intra-Topic Rsp**: Intra-topic self-resampling. **SR**: Selective replacement. **Both/Pref**: Performing selective replacement for both preferred and rejected responses or only for preferred responses while using original responses as rejected counterparts. **In-Ctx**: In-context rewriting for selective replacement. Here, we choose **(2a)** as our baseline, and any difference in other variants are emphasized in color.

| | Alternative Generate | | | Preference Curation | | ObjHal | | AMBER | |
|---|---|---|---|---|---|---|---|---|---|
| | Multi-Res. | Decompose | Intra-Topic Rsp. | Strategy | In-Ctx. | CH$_s \downarrow$ | CH$_i \downarrow$ | Acc. ↑ | F1 ↑ |
| **(2a)** | ✓ | ✓ | ✓ | SR+Both+Greedy | ✓ | **5.9** | **3.1** | **82.1** | **87.0** |
| **(2b)** | ✓ | ✓ | ✓ | SR+Both+Random | ✓ | 29.7 | 13.1 | 78.9 | 84.1 |
| **(2c)** | ✓ | ✓ | ✓ | SR+Pref+Greedy | ✓ | 6.4 | 3.2 | 76.8 | 84.7 |
| **(2d)** | ✓ | ✓ | ✓ | SR+Both+Greedy | ✗ | 35.5 | 20.1 | 80.1 | 84.4 |
| **(2e)** | ✗ | ✓ | ✓ | SR+Both+Greedy | ✓ | 7.2 | 4.0 | 80.2 | 86.4 |
| **(2f)** | ✓ | ✓ | ✗ | SR+Both+Greedy | ✓ | 12.6 | 6.6 | 77.9 | 84.9 |
| **(2g)** | ✓ | ✓ | – | Ranking | – | 9.7 | 4.8 | 80.9 | 85.9 |
| **(2h)** | ✓ | ✗ | – | Ranking | – | 25.5 | 12.0 | 73.5 | 82.8 |
| **(2i)** | ✗ | ✗ | – | Rewrite+Pref | – | 11.3 | 9.8 | 79.3 | 84.5 |
| **(2j)** | ✗ | ✗ | – | Rewrite+Both | – | 15.6 | 12.4 | 78.4 | 83.6 |

**Evaluation Benchmarks.** We assess visual hallucinations mitigation on Object HalBench [54], MMHal-Bench [15], AMBER [55] (discriminative part), RefoMB [17] and POPE [9]. We assess general capabilities on LLaVA-Bench [2] (in-the-wild) and MMStar [56].

## 4.2 Main Results

Results in Table 1 demonstrate TPR significantly mitigates visual hallucinations in VLMs across several benchmarks. The proposed TPR, in both its evaluated variants, achieves leading performance in mitigating visual hallucinations when applied to LLaVA-1.5-7B. Notably, our TPR-CL variant establishes new state-of-the-art results on several hallucination benchmarks, reducing hallucinations by ∼93% on Object-HalBench and ∼41% on MMHal-Bench. This substantial improvements can be consistently observed across other challenging hallucination benchmarks such as AMBER and RefoMB, underscoring the efficacy of our proposed design. Moreover, models fine-tuned using TPR-curated data not only maintain but sometimes enhance the base model's performance on general capability benchmarks. This outcome indicates that TPR can effectively suppresses visual hallucinations without compromising general VLM capabilities.

The introduction of a curriculum learning strategy (TPR-CL) consistently yields superior performance compared to the greedy variant across all evaluated metrics. By progressively exposing the model to more difficult examples according to a curricular scheduler, TPR-CL compels it to discern fine-grained details, complex contextual relationships, and subtle inconsistencies that often underpin persistent and challenging visual hallucinations. Crucially, TPR's unique capability to exercise fine-grained control over reward gap configuration is the key to effectively implementing curriculum learning strategy, thereby cultivating a more robust policy and delivering the state-of-the-art hallucination mitigation demonstrated by TPR-CL.

## 4.3 Ablation Studies

To investigate how TPR optimizes the reward gap configuration by enabling deliberate control over underlying data characteristics within preference pairs, we conduct a series of ablation studies. As presented in Table 2, our investigations focus on addressing the following questions: **(1)** How do the core components of TPR influence the quality and data characteristics of the resulting preference pairs, which directly shape the individual reward gaps? **(2)** How does the selective replacement mechanism in TPR impact final performance of the learned policy, when compared against other preference curation approaches, such as ranking-based or rewriting-based methods? **(3)** How to construct a high-quality rejected response $y_l$ in preference pair? Is it more effective to employ external "black-box" rewriters common in rewriting-based methods or to utilize model's internally resampled candidates, as TPR does? **(4)** From a broader perspective, what is the data efficiency of preference data curated by TPR, particularly when comparing its performance with that of other methods across

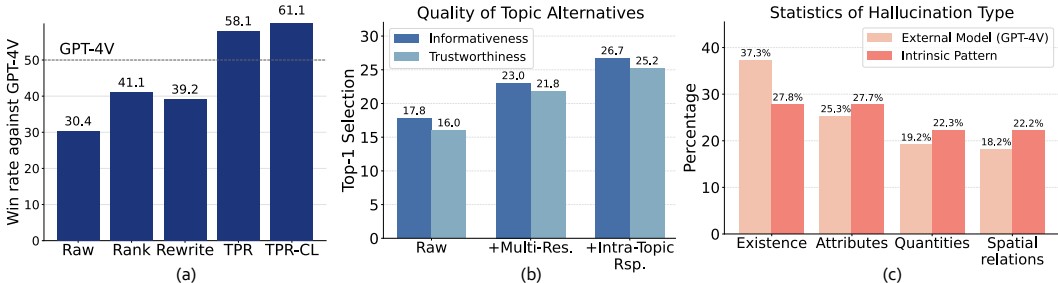

Figure 4: **(a) Quality of Overall Constructed Responses.** We compare responses generated from different preference curation strategies against strong "Ground Truth" responses from GPT-4V, where win-rates exceeding 50% indicate superior to GPT-4V outputs. For a fair comparison, we use LLaVA-NeXT-34B as the labeler or rewriter across all strategies, including "+Rank", "+Rewrite" and "+TPR(-CL)". **(b) Quality of Topic Alternatives.** The numbers on the bars indicate the top-1 selections made by GPT-4V in term of informativeness and trustworthiness. **(c) Hallucination Types.** Hallucinations introduced by external rewriters differ significantly from model's own failure modes.

varying data volumes? For computational efficiency, these ablation studies are conducted using a subset of 8,000 curated preference data instances. More ablations are provided in the Appendix G, including using different model architectures, using the reference model itself as labeler, and *etc.*

**Topic Alternatives Generation.** We conduct ablations on different approaches for generating topic alternatives. Results in Table 2 reveal that employing multiple initial candidate responses for decomposition and subsequently performing intra-topic self-resampling proves highly beneficial (exp **2a** *vs.* **2e/f**). It facilitates better coverage of edge cases, such as rarely mentioned attributes, and ultimately yields a more diverse and comprehensive set of alternatives. Such diversity is crucial for sculpting more sophisticated reward gap configuration during data curation. To quantify these aspects, Figure 4 (b) provides a direct comparison, evaluating the underlying data characteristics like informativeness and trustworthiness of these generated alternatives, based on GPT-4V reviews.

**Preference Curation Approaches.** We conduct ablations to analyze the impact of specific design choices within the preference curation process, as detailed in Table 2. Deviations from our base TPR variant (exp **2a**), such as employing random selection for replacement (exp **2b**), applying selective replacement to only one side of preference pair (exp **2c**, or omitting in-context rewriting (exp **2d**), all degrade final model performance. These findings highlight that the purposeful, symmetric replacement for both preferred and rejected responses, coupled with in-context rewriting, is crucial for crafting reward gaps that provide strong and effective learning signals for DPO.

Moreover, TPR's core mechanism of selective topic replacement demonstrably outperforms other preference curation strategies, such as ranking-based or rewriting-based methods (exp **2a** *vs.* **2g/h/i/j**). These performance advantages are further substantiated by quality assessments present in Figure 4 (a). In this evaluation, responses generated by different curation strategies are compared against strong "ground truth" (GT) responses from GPT-4V. We calculate one-vs.-one win-rates to quantify the quality of the generated responses, where win-rates exceeding 50% indicate that the respective preference curation strategy generates responses superior to those of GPT-4V. Additionally, more qualitative case studies are provided in Appendix E, offering an intuitive understanding of the quality improvements achieved by TPR. Collectively, all these result confirm that TPR's meticulous, topic-focused curation not only leads to better performance but also genuinely enhances the intrinsic quality of the preference data for robust VLM alignment.

**Rejected Response** $y_l$**.** We conduct ablations to assess how different $y_l$ curation approaches influence the nature of the resulting $y_l$, particularly its fidelity in incorporating hallucinations that reflect the base model's genuine failure modes. Some rewriting-based methods [26] employ powerful external models, such as GPT-4V, to modify responses, intending to create suitably "negative" $y_l$ instances. However, a critical concern is that the hallucination patterns introduced by these external rewriters may deviate from genuine deficiencies of the model that we aim to align. As illustrated in Figure 4 (c), the distribution of hallucination types in $y_l$ from these methods can differ significantly from model's intrinsic hallucination patterns. For instance, an external rewriter might introduce a more imbalanced distribution of various hallucination types, while the model's own intrinsic hallucination

patterns exhibit a relatively uniform distribution. This discrepancy results in a mismatch between the curated $y_l$ and model's actual deficiencies, thereby impairing alignment efficiency (exp **2a** *vs.* **2i** *vs.* **2j**) because the model is not optimally trained against its specific hallucinatory tendencies. Conversely, TPR performs intra-topic replacement on model-generated response template, ensuring that $y_l$ more accurately mirror the model's intrinsic failure distribution.

**Data Efficiency.** Building on the findings that TPR yields higher quality topic candidates and overall responses, we further investigate its data efficiency, as illustrated in Figure 1 (b). While RLHF-V [12], leveraging human annotation, demonstrates notable initial efficiency by achieving low hallucination rates with only 1.4k data, its prohibitive costs inherently limit scalability. In contrast, TPR, through its automated data curation process, facilitates a rapid reduction in hallucination levels when scaling from 4k to 20k data, ultimately surpassing human-annotated data in both performance and cost-effectiveness. Other automated methods generally exhibit inferior data efficiency. For instance, RLAIF-V [17], despite appearing competitive with 22k data volume, actually necessitates multiple rounds of data re-generation and retraining (effectively equivalent to an 88k data processing effort), significantly inflating its computational cost compared to TPR. These observations underscore how the enhanced data quality, achieved through TPR's systematic optimization of reward gap configuration, directly contributes to superior data efficiency.

## 5 Limitations and Future Prospects

Our work presents a promising direction for mitigating hallucinations, yet it also has limitations that open avenues for future research.

**Complexity of TPR Pipeline.** The TPR involves several sequential steps, including response decomposition, topic clustering, self-resampling, and in-context rewriting. While each stage is integral to ensuring high-quality data curation, their combination increases the overall complexity and computational overhead. In future work, we are committed to improving the framework's efficiency by simplifying the pipeline, *e.g.*, by investigating the necessity of the clustering step, to further enhance its throughput and reduce computational demands.

**Explore Optimal Data Curation Dimensions.** While our curriculum learning strategy in TPR-CL demonstrates one effective way to leverage TPR's control capabilities, the potential for optimizing data curation extends further. Investigating more advanced data curation dimensions, predicated on TPR's flexible control over semantic details, remains a rich and promising direction for enhancing VLM robustness against more challenging and complex hallucinations.

**Hallucinations in Multimodal Reasoning Tasks.** A critical next step is to extend these mitigation strategies to address hallucinations in complex multimodal reasoning tasks [61, 62, 63]. TPR is currently effective at reducing perceptual hallucinations, but adapting its topic-level control to handle errors in logical or causal reasoning chains presents a significant and valuable challenge for future exploration.

## 6 Conclusion

In this work, we introduce TPR, a novel paradigm designed for robust VLM alignment. By systematically controlling the reward gap configuration through topic-level alternatives generation and selective topic rewriting, TPR facilitates the construction of high-quality preference pairs with precisely controlled characteristics. This meticulous data curation is pivotal for generating effective training signals that significantly enhance model performance in mitigating hallucinations. Comprehensive experiments demonstrate that TPR, particularly when augmented with our proposed curriculum learning strategy (TPR-CL), achieves state-of-the-art performance on several hallucination benchmarks, significantly reducing hallucinatory outputs from the reference model.

## Acknowledgments and Disclosure of Funding

This work was supported by National Natural Science Foundation of China (62132001), the Shanghai Artificial Intelligence Laboratory, and the Fundamental Research Funds for the Central Universities.

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

# A Mathematical Grounding for Reward Gap Optimization

In this section, we provide a mathematical justification for the systematic optimization of the reward gap, connecting the data curation strategies introduced in Section 3 to the underlying learning dynamics of Direct Preference Optimization (DPO) [20].

## A.1 DPO Loss Function and Gradient Analysis

The DPO loss function is defined as:

$$\mathcal{L}_{\text{DPO}}(\pi_\theta; \pi_{\text{ref}}) = -\mathbb{E}_{(x,y_w,y_l)\sim\mathcal{D}} \left[ \log \sigma \left( \beta \log \frac{\pi_\theta(y_w|x)}{\pi_{\text{ref}}(y_w|x)} - \beta \log \frac{\pi_\theta(y_l|x)}{\pi_{\text{ref}}(y_l|x)} \right) \right] \tag{2}$$

where $\sigma$ is the sigmoid function and $\beta$ is a hyperparameter controlling the deviation from the reference model $\pi_{\text{ref}}$. For a single preference pair, let $M_{\pi_\theta}$ be the policy model's estimated log-probability ratio (or estimated reward gap):

$$M_{\pi_\theta} = \log \frac{\pi_\theta(y_w|x)}{\pi_{\text{ref}}(y_w|x)} - \log \frac{\pi_\theta(y_l|x)}{\pi_{\text{ref}}(y_l|x)} \tag{3}$$

The gradient of the loss with respect to the model parameters $\theta$ is then given by:

$$\nabla_\theta \mathcal{L}_{\text{DPO}} = -\beta \cdot \sigma(-\beta M_{\pi_\theta}) \left[ \nabla_\theta \log \pi_\theta(y_w|x) - \nabla_\theta \log \pi_\theta(y_l|x) \right] \tag{4}$$

The term $[\nabla_\theta \log \pi_\theta(y_w|x) - \nabla_\theta \log \pi_\theta(y_l|x)]$ determines the **direction** of the update, encouraging the model to increase the likelihood of the preferred response $y_w$ and decrease that of the rejected response $y_l$. The term $\sigma(-\beta M_{\pi_\theta})$ determines the **magnitude** of the gradient update. Our analysis focuses on this magnitude term.

## A.2 True Reward Gap in Learning Dynamics

Let $r^*(y, x)$ represent an oracle or true reward function that perfectly captures the desired behavior (*e.g.*, factual accuracy). For any preference pair $(y_w, y_l)$ we curate, we can define the **true reward gap** as:

$$\Delta r^* = r^*(y_w, x) - r^*(y_l, x) \tag{5}$$

This $\Delta r^*$ is an intrinsic property of the data pair that reflects its difficulty. A large $\Delta r^*$ signifies an "easy" pair (*e.g.*, a factually correct response vs. a blatant hallucination), while a small $\Delta r^*$ signifies a "hard" pair (*e.g.*, a correct response vs. a subtly flawed one).

The curriculum learning strategy in TPR (see Section 3.4) systematically manipulates $\Delta r^*$ to optimize the learning trajectory. We analyze the impact of this manipulation on the model's estimated gap $M_{\pi_\theta}$ and, consequently, the gradient magnitude.

**Training with High Reward Gaps (Warm-Up Stage).** Initially, the model is trained on pairs with a large true reward gap ($\Delta r^* \gg 0$). The model quickly learns to differentiate these "easy" examples, causing its estimated gap $M_{\pi_\theta}$ to become large and positive. As $M_{\pi_\theta} \to \infty$, the gradient magnitude term $\sigma(-\beta M_{\pi_\theta}) \to 0$. The learning signal for these easy examples diminishes, indicating the model has mastered them.

**Training with Low Reward Gaps (Hard-Mining Stage).** After the warm-up stage, TPR introduces "hard" pairs where $y_l$ is subtly incorrect, corresponding to a small true reward gap ($\Delta r^* \to 0$). For these challenging pairs, the model initially struggles to distinguish $y_w$ from $y_l$, resulting in an estimated gap $M_{\pi_\theta} \approx 0$. When $M_{\pi_\theta}$ is near zero, the gradient magnitude $\sigma(-\beta M_{\pi_\theta})$ is at its maximum ($\approx 0.5$). This creates a strong learning signal, forcing the model to focus on the fine-grained details it was previously ignoring and refine its decision boundary. In our curriculum strategy, by progressively reducing the true reward gap in the training data, we keep the model training in a high-gradient magnitude regime for increasingly difficult problems. This leads to a more robust and fine-tuned policy against challenging hallucinations.

## A.3 Comparison with Adjusting the Hyperparameter $\beta$

Based on Equation (4), an alternative way to amplify the learning signal when $M_{\pi_\theta}$ is large would be to dynamically decrease the hyperparameter $\beta$. However, this approach has potential drawbacks. The

parameter $\beta$ is fundamentally designed to control the strength of the KL-divergence penalty between the policy $\pi_\theta$ and the reference $\pi_{\text{ref}}$. Aggressively lowering $\beta$ to force learning on hard examples could weaken this constraint, potentially leading to unforeseen distribution shifts away from the reference model. In contrast, TPR's approach of manipulating the data's intrinsic reward gap $\Delta r^*$ is more direct. By using the reference model itself for every creative step (resampling, rewriting), TPR ensures that even the "hard" preference pairs remain grounded in the model's own failure modes and capabilities, mitigating the risk of drastic policy divergence.

## B  Topic Clustering

To determine if two semantic units, $u_{m,n}$ and $u_{p,q}$, belong to the same topic $c$, we evaluate their textual consistency and visual correlation.

**Textual Consistency.** We assess textual consistency by querying the reference model $\pi_{\text{ref}}$ to determine if $u_{m,n}$ and $u_{p,q}$ describe the same core idea. The model is prompted as follows:

$$p_{\text{text}}(u_{m,n}, u_{p,q}) = \pi_{\text{ref}}(u_{m,n}, u_{p,q}|\ \textit{"Are}\ \{u_{m,n}\}\ \textit{and}\ \{u_{p,q}\}\ \textit{describing}$$
$$\textit{the same topic? Please answer Yes or No.")} \tag{6}$$

$p_{\text{text}}$ is considered `True` if $\pi_{\text{ref}}$ outputs Yes, and `False` otherwise.

**Visual Correlation.** As illustrated in Figure 5, we also assess whether two semantic units visually correspond to similar regions within the input image. To achieve this, we utilize the vision tower in VLM, specifically CLIP [44], to extract pooled text embeddings for each candidate semantic unit $u$ and image embeddings for each vision token $v$. We then compute the similarity $\text{Sim}(u, v)$ between these text and image embeddings. Semantic units are considered visually correlated if their respective similarity vectors, when compared to the image embeddings, exhibit a high correlation:

$$p_{\text{vis}}(u_{m,n}, u_{p,q}) = \texttt{Correlation}\Big[\texttt{Sim}(u_{m,n}, v), \texttt{Sim}(u_{p,q}, v)\Big] > \tau_{\text{vis}} \tag{7}$$

Here, `Correlation`$(\cdot)$, implemented using Pearson correlation, measures the correlation between these two similarity vectors. $p_{\text{vis}}$ is considered `True` if the pre-defined threshold $\tau_{\text{vis}}$ is satisfied, and `False` otherwise. This visual correlation mechanism enables the distinction of multiple entities or aspects within an image, even when their textual descriptions are similar, thereby categorizing them into separate topics.

**Greedy Clustering.** Semantic units are considered to belong to the same topic if they exhibit both textual consistency (as defined by $p_{\text{text}}$) and visual correlation (as defined by $p_{\text{vis}}$). Specifically, we adopt an approach analogous to the Louvain method [45]. Louvain method is a greedy, iterative algorithm designed to optimize modularity, a metric quantifying the density of connections within communities relative to connections between them. The process begins by assigning each semantic unit to its own distinct community. Subsequently, for each unit, the algorithm evaluates whether moving it to a neighboring community would increase the overall modularity. This step is repeated iteratively until no individual move can further enhance modularity. This process, consequently, results in a partitioning of the semantic units into distinct topic clusters.

**Sensitivity Analysis.** We conducted a sensitivity analysis to evaluate the impact of the hyperparameter $\tau_{\text{vis}}$ on the final performance. As detailed in Table 3, the results indicate that employing a stricter threshold for matching candidate semantic units generally leads to improved performance. Consequently, based on this analysis, we set $\tau_{\text{vis}} = 0.9$ for all experimental setups.

## C  Curriculum Learning

To progressively refine model's ability to discern subtle errors, TPR-CL (**C**urriculum **L**earning) employs an iterative alignment approach. It incorporates a selective replacement mechanism for semantic units, guided by a curriculum. This curriculum dictates the selection of alternative semantic units and subsequently guides the construction of rejected responses $y_l$ paired with preferred ones $y_w$.

**Implementation Details.** The TPR-CL process is divided into multiple iterations. In each iteration, preference data is generated using a distinct subset of the overall data sources (as detailed in Section 4.1, containing images $I$ and prompts $x$). This newly generated preference data is then used to

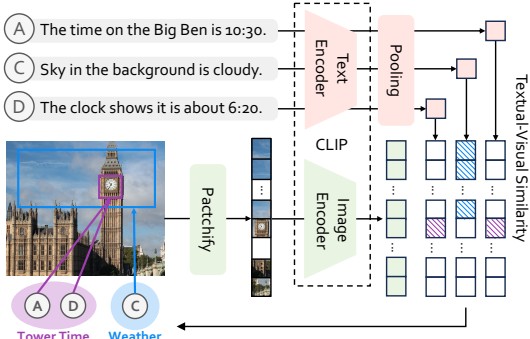

Figure 5: **Visual Correlation.**

Table 3: **Ablation studies on topic clustering.** We ablate the impact of $p_{\text{text}}$ and $p_{\text{vis}}$ in topic clustering, and conduct a sensitivity analysis of $\tau_{\text{vis}}$, used in visual correlation step of topic clustering.

| | Condition | $\tau_{\text{vis}}$ | ObjHal | | AMBER | |
|---|---|---|---|---|---|---|
| | | | $\text{CH}_s\downarrow$ | $\text{CH}_i\downarrow$ | Acc. $\uparrow$ | F1 $\uparrow$ |
| **(3a)** | $p_{\text{text}}$ | - | 13.4 | 6.8 | 80.1 | 85.9 |
| **(3b)** | $p_{\text{vis}}$ | 0.9 | 13.0 | 7.3 | 80.1 | 86.1 |
| **(3c)** | | 0.6 | 8.3 | 4.1 | 80.8 | 86.1 |
| **(3d)** | $p_{\text{text}} + p_{\text{vis}}$ | 0.8 | 7.2 | 3.5 | **82.5** | 86.9 |
| **(3e)** | | 0.9 | **5.9** | **3.1** | 82.1 | **87.0** |
| **(3f)** | | 0.95 | 6.7 | 3.5 | 81.8 | 86.8 |

fine-tune the policy model $\pi_\theta$. The aligned policy model resulting from one iteration serves as the reference model for the subsequent iteration. Note that unlike methods such as RLAIF-V [17] which re-generates preferences using the entire data sources in each of its multiple iterations (a total generation effort equivalent to $N_{\text{iterations}} \times |\mathcal{D}|$), TPR-CL is designed so that preference data for any given subset of the data sources is generated only once throughout the entire multi-iteration process (a total generation effort equivalent to $|\mathcal{D}|$).

During all iterations, preferred responses $y_w$ are constructed by consistently selecting the highest-scored alternative semantic units. The strategy for constructing rejected responses $y_l$ varies according to the curriculum: **(1)** In first 60% iterations (denoted as "Warm-Up" stage), $y_l$ is constructed by replacing semantic units with the lowest-scored alternatives from the candidate pool, encouraging faster initial convergence of the policy model. **(2)** In remaining 40% iterations (denoted as "Hard-Mining" stage), the difficulty of the negative examples is gradually increased to further refine the model. For iterations spanning 60% to 80% of the total, alternatives for $y_l$ are selected from those scoring in the bottom 20% of the candidate pool. For the final 80% to 100% iterations, alternatives for $y_l$ are selected from those scoring between the bottom 20% and bottom 40% of the candidate pool. As the policy model $\pi_\theta$ becomes more capable through these iterations, the progressively increasing score threshold for negative examples challenges it to make finer distinctions.

**Effectiveness of Hard Mining.** To investigate the impact of curriculum learning on model's ability to avoid subtle errors, we conduct a quantitative analysis comparing effectiveness in reducing hallucination across different types of varying difficulties. The RefoMB benchmark [17] is used for this ablation, as its detailed categorizations allow us to pinpoint the benefits of curriculum learning more precisely. The results in Figure 6 clearly demonstrate the progressive improvements achieved through our proposed TPR method and curriculum learning strategy. On challenging hallucinations for baseline LLaVA-1.5 model, *i.e.*, "Quantities" (16.7%) and "Spatial Relations" (14.3%), the introduction of curriculum learning in TPR-CL yields further substantial improvements, providing an additional +20.8/+35.7 point increase over its greedy variant. For "Existence" and "Attributes", where the greedy variant already performed strongly, TPR-CL still offers valuable refinements, with additional gains of +5.3 and +8.3 points respectively. These findings underscore the effectiveness of the hard mining component within our curriculum learning strategy. This is especially crucial for tackling more subtle and complex hallucinations related to quantities and spatial relationships, where the baseline model and even the greedy TPR approach show limitations.

## D   Computational Cost

To evaluate the practical viability of TPR, we conduct a comprehensive computational cost analysis. As illustrated in Figure 7, TPR demonstrates the most favorable efficiency trajectory, achieving a rapid and substantial reduction in hallucinations at a minimal cost. This superior efficiency is rooted in our data generation pipeline, with a detailed breakdown provided in Table 4. The process of generating our 20k preference dataset on 8 NVIDIA A100 GPUs requires 71.05 hours, a time significantly reduced to just 26.04 hours with the integration of the vLLM [66] inference engine. This culminates in a highly efficient rate of 4.7 GPU-seconds per generated pair.

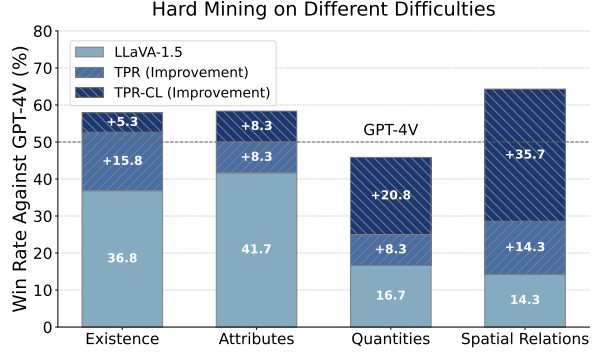

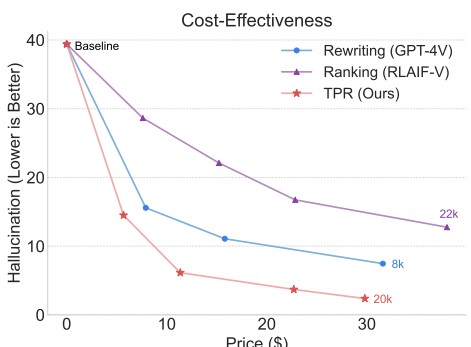

Figure 6: **Effectiveness of Hard Mining.** We compare TPR with curriculum learning strategies against its greedy variant and LLaVA-1.5 baseline. Experiments are conducted on RefoMB benchmark using the same training setup for fair comparison.

Figure 7: **Cost-Effectiveness.** We compare the cost-effectiveness of TPR against rewriting- and ranking-based methods. The numbers next to the lines indicate the total data generation efforts.

Table 4: **Computational Cost Breakdown.** We provide the total computational cost of TPR generating full 20k dataset in each stage. vLLM is applied as inference engine in TPR for acceleration. All results are obtained using 8 NVIDIA A100 GPUs.

| Stage | TPR | TPR w/ vLLM |
|---|---|---|
| Response Generation | 10.27h | 1.36h |
| Decomposition | 8.91h | 3.57h |
| Wh-question Convertion | 7.95h | 3.17h |
| Self-resampling | 7.48h | 2.96h |
| Topic Cluster | 10.84h | 6.71h |
| Scoring & Ranking | 23.43h | 7.42h |
| In-context Rewriting | 2.17h | 0.85h |
| **Total** | **71.05h** | **26.04h** |

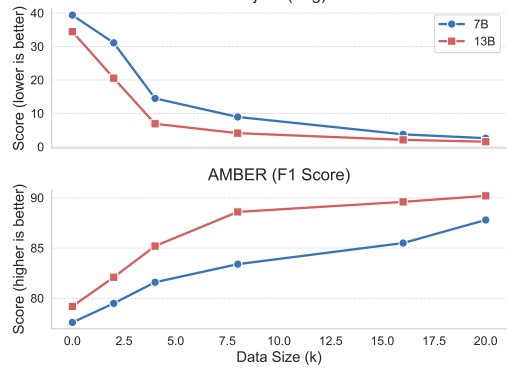

Figure 8: **Larger Base Model.** We provide the results of 7B/13B base model on 2k~20k data.

When compared with methods relying on proprietary APIs, TPR's advantage is pronounced. As shown in Figure 7, while a rewriting approach using GPT-4V yields a significant performance improvement, it comes at a much higher price. TPR achieves a comparable level of hallucination reduction at approximately one-third of the estimated API cost required by the GPT-4V approach (roughly $10 for TPR *vs.* over $30 for GPT-4V). Furthermore, TPR is also more cost-effective than other open-source alternatives like RLAIF-V. Under the same hardware conditions, generating a 22k dataset with the official RLAIF-V implementation takes approximately 66 hours. Although RLAIF-V involves fewer distinct stages, its process is dominated by a computationally expensive scoring phase using a 34B parameter labeler model. More critically, its iterative refinement strategy introduces significant overhead: the policy model must be repeatedly retrained between data generation rounds, a resource-intensive process that cannot be accelerated by inference engines like vLLM. This inherent inefficiency explains the gentler decline in its performance-to-cost curve shown in Figure 7.

In summary, these findings clearly indicate that TPR not only achieves state-of-the-art hallucination reduction but does so with significantly greater computational and financial efficiency than existing methods, highlighting its practical value for large-scale VLM alignment.

# E  Qualitative Case Studies

We provide qualitative case studies to illustrate two key aspects: **(1)** The differences between preference pairs generated by TPR and those generated by its curriculum learning variant, TPR-CL. See Figure 9 and 10. **(2)** The enhanced quality of responses from the policy model fine-tuned with

TPR-curated data, particularly when compared against both the LLaVA-1.5-7B baseline and the larger LLaVA-NeXT-34B model. See Figure 11 and 12.

# F  More Implementation Details

## F.1  Data Curation

**Hyper-Parameters.** For preference data curation, we utilize instruction prompt $x$ from RLHF-V [12] to guide the reference model $\pi_{\text{ref}}$ in generating $M = 10$ candidate responses $\{y_1, y_2, \cdots, y_M\}$ for each image $I$. After these responses are decomposed into semantic units, $\pi_{\text{ref}}$ is queried with wh-questions derived from each unit to self-resample one additional candidate per topic. For the TPR-CL variant, 12,000 instances (60%) are constructed during "Warm-Up" stage, and the remaining 8,000 (40%) are constructed during "Hard-Mining" stage. For initial candidate responses sampling and intra-topic resampling, we follow RLHF-V [12] to set the parameters as: `temperature=0.7,top_p=0.95,do_sample=True`. Other parameters like `top_k` are left at their default values.

**Prompts.** Within the TPR paradigm, the reference model $\pi_{\text{ref}}$ is prompted to perform several key operations: candidate response generation, response decomposition, topic clustering, in-context rewriting and the conversion of semantic units into both wh-questions and yes-no questions. The specific prompts utilized to guide $\pi_{\text{ref}}$ for these tasks are detailed in Table 5 and Table 6.

## F.2  Evaluation

**Benchmarks.** We evaluate TPR on several benchmarks:

- **Object-HalBench.** Rohrbach, et al. [54] is designed for common object hallucinations in detailed image descriptions. We follow Yu, et al. [17, 12] to use 8 diverse prompts to improve the stability during evaluation. We report the CHAIR$_s$ (the percentage of hallucinatory responses) and CHAIR$_i$ (the percentage of hallucinated objects).

- **MMHal-Bench.** Sun, et al. [15] evaluates hallucinations and informativeness by using GPT-4 [28] to compare model outputs with human annotations.

- **AMBER.** Wang, et al. [55] evaluates the object existence, attributes and relations in the image description. We use discriminative part of AMBER for evaluation, and report the accuracy and F1 metric.

- **RefoMB.** Yu, et al. [17] consists of 120 images, each paired with 3 instructional annotations, and evaluates 8 fundamental competencies covering both hallucination and reasoning.

- **POPE.** Li, et al. [9] evaluates the object existence through querying the VLM with close-ended yes-no questions. Note that we use original prompts in POPE during evaluation for stability. We report the F1 score and accuracy on three different sampling strategies in POPE, *i.e.*, adversarial, popular and random sampling. We also report the overall F1 score.

- **LLaVA-Bench.** We use LLaVA-Bench [2] (in-the-wild) to evaluate VLMs in multimodal conversation, detailed descriptions and reasoning aspects. We report the overall score.

- **MMStar.** Chen, et al. [56] evaluates VLMs on 6 core capabilities and 18 specific aspects related to general capabilities. We report the overall score.

**Comparison Counterparts.** We compare our TPR with multiple RLHF/RLAIF methods:

- **LLaVA-RLHF.** Sun, et al. [15] first fine-tunes LLaVA [2] with manual-annotated instruction tuning datasets, *i.e.*, VQA-v2 [46], A-OKVQA [51] and Flickr30k [64], to enhance its general capabilities. Subsequently, it trains a reward model on 10k preference data derived from human feedback and applies PPO [65] on 72k factually augmented data for preference learning.

- **RLHF-V.** Yu, et al. [12] collects 1.4k fine-grained preference data in the form of segment-level corrections on hallucinations through manual annotation. It then aligns the VLMs using the proposed dense DPO.

Table 5: **Prompts Used in Topic-level Alternative Generation.** Here, {question}, {answer} and {sentence} are placeholders that will be replaced when inputted into the reference model.

---

**Candidate Response Generation (Modified from RLHF-V)**

We randomly select a following question to use as input for the reference model, prompting it to generate responses based on the image.

**### Questions:**
- What is the setting or environment in which the image takes place?
- Provide an intricate description of every entity in the image.
- Can you point out the details that make this image unique?
- What are the main elements in this image? Describe them thoroughly.
- Identify and describe each object in the image in detail.
- Analyze this art image, describing its spatial arrangement, interactive elements, and conceptual message.
- Detail the texts and other components in the image in depth, explaining their relevance to the overall picture.
- Look at the image and describe the celebrity's facial expressions, clothing, and any distinctive features.
- ...

- - - - - - - - - - - - - - - - - - - - - - - - - - - - - - - - - - - - - - - - - - - - - - - - - - - - - -

**Response Decomposition (Modified from RLAIF-V)**

You are an expert in extracting facts from the given question-answer pair for an image. Your task is to extract and rewrite the facts mentioned in the answers into self-contained sentences. Exclude opinions or subjective statements.

You should present your result in the following format:

**### Facts:**
- {Extracted fact 1}
- {Extracted fact 2}
- ...

**### Question-answer pair:**
Question: {question}
Answer: {answer}

- - - - - - - - - - - - - - - - - - - - - - - - - - - - - - - - - - - - - - - - - - - - - - - - - - - - - -

**Wh-Question Converting**

You are an expert at modifying a given declarative sentence into a wh-question sentence. Your task is to modify the given declarative sentences one by one into a wh-question form. Do not change tenses or add extra content.

You should present your result in the following format:

**### Converted questions:**
- {Converted question 1}
- {Converted question 2}
- ...

**### Declarative sentences:**
- {sentence 1}
- {sentence 2}
- ...

---

- **Silkie.** Li, et al. [24] adopts GPT-4V [1] to assess the responses generated by multiple VLMs regarding helpfulness, visual faithfulness and ethical considerations. It then applies DPO [20] to train Qwen-VL-Chat [4] over 80k GPT-4V preferences.

- **POVID.** Zhou, et al [26] emphasizes the importance of rejected responses and generates high-quality rejected responses by distorting the image and injecting additional hallucinations using GPT-4V. It then fine-tunes the LLaVA-1.5-7B with generated 17k preference data.

Table 6: **Prompts Used in Topic Selective Rewriting.** Here, {sentence}, {tip}, {question} and {answer} are placeholders that will be replaced when inputted into the reference model.

---

**Yes-no Question Converting**

You are an expert at modifying a given declarative sentence into a general question sentence. Your task is to modify the given declarative sentences one by one into a general question form. Do not change tenses or add extra content.

If the given declarative sentence contains not, no or negative meaning words, you need to check the modified general interrogative sentence to make sure that the generated general question sentence retains words with not, no or negative meaning words.

You should present your result in the following format:

### Converted questions:
- {Converted question 1}
- {Converted question 2}
- ...

### Declarative sentences:
- {sentence 1}
- {sentence 2}
- ...

- - - - - - - - - - - - - - - - - - - - - - - - - - - - - - - - - - - - - - - - - - - - - - - - -

**Textual Consistency Evaluation**

You are an expert at determining if the given two declarative sentences are consistent in textual semantics. Your task is to determine if the topic described in these two sentences are consistent. If you can confirm that two sentences are consistent, please output "consistent". Otherwise, output "unrelated".

### Declarative sentences:
- {sentence 1}
- {sentence 2}

- - - - - - - - - - - - - - - - - - - - - - - - - - - - - - - - - - - - - - - - - - - - - - - - -

**In-Context Rewriting**

You are an expert at modifying a declarative answer with several tips. Your task is to modify the original answer, which is used to answer the question, based on the image and the provided tips. The given tips will relate to a specific part of the original answer, and you should use the tips to overwrite the corresponding part. If there is a conflict between the tips and the image, remember to follow the tips first.

You should make minimal modifications and maintain style and format with the original answer. Only output the modified answer.

### Tips:
- {tip 1}
- {tip 2}
- ...

### Question-answer pair:
Question: {question}
Original Answer: {answer}

---

- **MFPO.** Jiang, et al. [57] introduces image-related rewards in preference data and constructs 1.4k image preference data upon RLHF-V. It then aligns the VLMs with proposed modality-fair preference optimization (MFPO).

- **AMP.** Zhang, et al. [19] designs an automated pipeline that generates multi-level preference data for multi-level comparison. It then uses 11k multi-level preference data to align VLMs with proposed multi-level DPO.

- **RLAIF-V.** Yu, et al. [17] adopts a divide-and-conquer strategy that determines the overall response score by aggregating the decomposed sub-response scores, mitigating the expensive

Table 7: **More Ablations on Self-Labeling and Different Model Architecture.**

| Model | Hallucination Benchmarks | | | | | | | | | General Benchmarks | |
|---|---|---|---|---|---|---|---|---|---|---|---|
| | ObjHal | | MMHal | | AMBER | | RefoMB | | POPE | LLaVA-B | MMstar |
| | CH$_s$ ↓ | CH$_i$ ↓ | Score ↑ | Hall. ↓ | Acc. ↑ | F1 ↑ | Trust. ↑ | Win. ↑ | F1 ↑ | Overall ↑ | Overall ↑ |
| LLaVA-1.5-7B | 53.6 | 25.2 | 2.36 | 51.0 | 73.5 | 77.6 | 30.8 | 12.1 | 85.9 | 59.7 | 30.3 |
| + TPR-SL | **5.8** | **3.0** | **2.67** | **44.8** | **81.7** | **86.7** | **53.5** | **30.3** | **86.1** | **73.7** | **32.8** |
| Qwen-VL-2B [4] | 42.4 | 36.3 | 2.85 | 47.9 | 68.3 | 81.0 | 48.5 | 20.7 | 86.5 | 83.2 | 29.1 |
| + Naive RLAIF | 36.5 | 31.7 | 2.81 | 49.0 | 70.9 | 81.8 | 52.5 | 23.7 | 87.0 | 83.2 | 30.3 |
| + TPR-SL-2B | **19.5** | **13.7** | **2.98** | **43.8** | **74.3** | **84.3** | **56.6** | **28.3** | **87.1** | **83.8** | **31.2** |

demand for ultra-large proprietary VLMs. It generates 28k preference data for preference learning with proposed iterative DPO.

- **HSA-DPO.** Xiao, et al. [27] first trains a hallucination detection model on hallucination datasets built by GPT-4V, and then follows a detect-then-rewrite pipeline to construct 6k preference data. It then aligns VLMs with proposed hallucination severity-aware DPO.

# G More Ablation Studies

**Larger Base Model.** To investigate the scalability of our method, we applied TPR-CL to both 7B and 13B variants of the base model across data scales from 2k to 20k. As illustrated in Figure 8, the 13B model consistently outperforms its 7B counterpart at every data point, achieving better hallucination scores in both ObjHal [54] and AMBER [55] benchmarks. This result confirms that the performance gains from TPR are complementary to the inherent capabilities of larger models, highlighting its strong scalability.

**Self-Labeling: Using the Reference Model Itself as Labeler.** The approach of scoring fine-grained semantic units within our intra-topic ranking mechanism potentially alleviates the demand for an exceptionally capable labeler model [17]. This prompted an exploration into the effectiveness of using the reference model itself for scoring and ranking after self-resampling, thereby investigating the boundaries of model self-improvement. Accordingly, we introduce TPR-SL (**S**elf-**L**abeling), a variant of TPR that leverages the base LLaVA-1.5-7B model [6] as its own preference labeler, instead of a more powerful one like LLaVA-NeXT-34B [3] used in our main experiments. The performance of TPR-SL compared to the LLaVA-1.5-7B baseline is presented in Table 7. Specifically, TPR-SL achieves substantial improvements over the baseline LLaVA-1.5-7B across both hallucination mitigation and general VLM capabilities. These findings are significant as they highlight the effectiveness of TPR in a self-labeling scenario. It validates a practical and efficient paradigm for VLM self-improvement that minimizes reliance on human annotations or powerful external models.

**Generalization on Different Model Architecture.** To verify the generalizability of the TPR paradigm, we apply it to a different model architecture, Qwen-VL-2B [4]. Specifically, Qwen-VL-2B serves as the reference model for both generating topic-level alternatives and subsequent selective replacement, with intra-topic ranking performed by Qwen-VL-2B itself (self-labeling). The preference data curated through this process is then used to fine-tune Qwen-VL-2B. As presented in Table 7, we compare this aligned policy model against the original Qwen-VL-2B baseline and the same base model fine-tuned using a naïve RLAIF approach (which employs an external LLaVA-NeXT-34B model for ranking). The results clearly demonstrate the effectiveness and generalizability of applying TPR paradigm, even in a self-labeling setup, to improve performance on both hallucination and general benchmarks. This aligns with the conclusions outlined in our main results (see Section 4.2). Therefore, it is affirmed that the TPR paradigm is not confined to a specific model architecture like LLaVA. Its core data curation design can be effectively adapted to other models such as Qwen-VL-2B, leading to significant reductions in hallucinations while maintaining or improving general VLM capabilities.

# H  Broader Impacts

The research presented in this paper aims to enhance the reliability of Vision Language Models (VLMs). By significantly reducing visual hallucinations, TPR leads to more factually accurate and contextually coherent outputs, which in turn can bolster user trust. These improvements are pivotal for unlocking safer and more effective VLM applications across diverse domains, from assistive technologies to educational tools, transforming these models into more reliable instruments. However, the deployment of this technology require careful consideration of potential societal impacts and technical trade-offs.

**Potential for Misuse.** A primary concern is that the synthetic "rejected" responses ($y_l$) generated by TPR, which are designed to contain plausible hallucinations, could be isolated and misused for creating disinformation if taken out of context. To mitigate this risk, we are committed to responsible asset release. Any public distribution of our curated preference dataset will be governed by a strict, research-focused license prohibiting malicious use. Furthermore, the dataset will be accompanied by comprehensive documentation that explicitly warns users that the negative examples are synthetic, generated solely for model alignment, and are factually incorrect and unsuitable for any other application.

**Environmental and Computational Cost.** Large-scale data curation and model alignment inevitably have an environmental cost due to the required computational resources. As detailed in our analysis (see Section D), TPR is applied during the post-training alignment phase, making its environmental impact substantially lower than that of pre-training a VLM from scratch. Crucially, our work emphasizes data efficiency. By generating high-quality preference pairs that provide strong learning signals, TPR achieves state-of-the-art performance with less data than many competing methods, thereby promoting a smaller environmental footprint for VLM alignment.

**Technical Safeguards.** Besides, a key technical consideration when applying TPR is ensuring that targeted improvements in factuality do not compromise the model's broader general capabilities. Although our experiments (see Table 1) show that our approach maintains or even enhances general performance, achieving this balance requires ongoing research and careful validation.

Successfully navigating these challenges is essential for the responsible development and widespread beneficial adoption of advanced VLM technology.

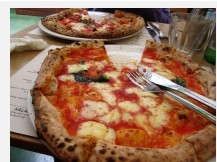

**What are the main elements in this image?**

**Raw Response:**
In the image, there is a dining table with two pizzas prominently placed on it. **One pizza is located towards the left side of the table, while the other pizza is positioned more towards the center**. In addition to the pizzas, there is **a fork and a knife** on the table, ready for use. **There are two people in the scene, with one person located on the left side and the other person on the right side of the table. A cup can be seen on the right side of the table**, **and a bowl is placed near the top part of the table.** A bottle is also visible on the right side, adding to the variety of items on the table.

**Preferred Response:**
In the image, there is a dining table with two pizzas prominently placed on it. **One pizza is located towards the left side of the table, while the other pizza is positioned more towards the center.** In addition to the pizzas, there is **a fork and a knife** on the table, ready for use. **A cup can be seen on the right side of the table,** and a bottle is visible on the right side, adding to the variety of items on the table.

- - - - - - - - - - - - - - - - - - - - - - - - - - - - - - - - - - - - - - - - - - - - - - - - - - - - - - - - - -

**Rejected Response (Greedy Strategy):**
The main elements in this image are a dining table with two pizzas placed on top of paper plates. **One pizza is larger and takes up most of the table surface, while the other pizza is smaller and located towards the left side of the table**. There is a fork, a knife, a cup, **and a bowl** on the table, indicating that the table is set for a meal. **Two people can be spotted in the scene, with one person on the left side and the other person on the right side of the table.** Additionally, a bottle is visible on the right side of the table.

- - - - - - - - - - - - - - - - - - - - - - - - - - - - - - - - - - - - - - - - - - - - - - - - - - - - - - - - - -

**Rejected Response (Curriculum Strategy):**
In the image, there is a dining table with two pizzas prominently placed on it. **One pizza is located towards the left side of the table, while the other pizza is positioned more towards the center.** In addition to the pizzas, there is **a fork and a knife** on the table, ready for use. **A cup can be seen on the right side of the table**, **and a bowl is placed near the top part of the table.** A bottle is also visible on the right side, adding to the variety of items on the table.

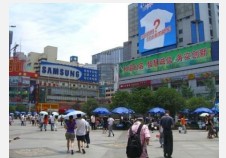

**Provide description of every entity in the image.**

**Raw Response:**
The image shows a large group of people walking outside of a crowded building, **with several individuals holding umbrellas to shield themselves from the elements**. **The crowd consists of people of various heights and positions**, many of them wearing backpacks. The bustling street is full of people engaged in their daily activities, while others are carrying handbags. The area has a lively atmosphere, with some individuals walking around, and others standing or sitting on benches. **The scene captures the essence of urban life and the diversity of people present in the city.**

**Preferred Response:**
The image shows a large group of people walking outside of a crowded building. **The crowd consists of people of various heights and positions**, many of them wearing backpacks. The bustling street is full of people engaged in their daily activities, while others are carrying handbags. The area has a lively atmosphere, with some individuals walking around, and others standing or sitting on benches. **The scene captures the essence of urban life and the diversity of people present in the city.**

- - - - - - - - - - - - - - - - - - - - - - - - - - - - - - - - - - - - - - - - - - - - - - - - - - - - - - - - - -

**Rejected Response (Greedy Strategy):**
The image depicts a lively city scene with a large group of people walking around in front of a crowded building. **Various individuals are carrying umbrellas to protect themselves from the elements**, and several backpacks are visible among the crowd. The atmosphere is bustling, with people engaged in their daily activities, walking, and standing or sitting on benches. **In total, there are 13 people in the image, including individuals near the left, center, and right sides of the scene. The presence of numerous umbrellas and backpacks suggests that it is raining.**

- - - - - - - - - - - - - - - - - - - - - - - - - - - - - - - - - - - - - - - - - - - - - - - - - - - - - - - - - -

**Rejected Response (Curriculum Strategy):**
The image shows a large group of people walking outside of a crowded building, **with several individuals holding umbrellas to shield themselves from the elements**. **The crowd consists of people of various heights and positions**, many of them wearing backpacks. The bustling street is full of people engaged in their daily activities, while others are carrying handbags. The area has a lively atmosphere, with some individuals walking around, and others standing or sitting on benches. **The scene captures the essence of urban life and the diversity of people present in the city.**

Figure 9: **Qualitative Results of Preferred and Rejected Responses Generated by TPR and TPR-CL. Correct answers** and **hallucinations** are highlighted in color respectively.

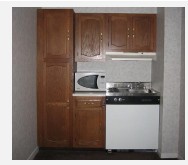

**Describe the elements in the image.**

**Raw Response:**
The image showcases a small kitchen with various appliances and furnishings. There is **a white dishwasher located towards the right side of the room**, and **a small microwave** situated **in the middle of the kitchen**. A sink can be seen **in the left corner of the kitchen**. **Numerous cabinets and cupboards** are present in the kitchen, providing ample storage space. **There is a refrigerator positioned in the middle of the kitchen, and a toaster and a bowl can be seen on a countertop**. The layout of the appliances creates an organized and functional kitchen space.

**Preferred Response:**
The image showcases a small kitchen with various appliances and furnishings. There is **a small microwave situated in the middle of the kitchen** and **a sink can be seen in the right side of the kitchen. Numerous cabinets and cupboards** are present in the kitchen, providing ample storage space. The layout of the appliances creates an organized and functional kitchen space.

- - - - - - - - - - - - - - - - - - - - - - - - - - - - - - - - - - - - - - - - - - - - - - - - - - - - - - - - - - - -

**Rejected Response (Greedy Strategy):**
The image showcases a small kitchen with various appliances and furnishings. There is **a white dishwasher located towards the right side of the room and a sink can be seen in the left corner of the kitchen. The cabinets are open** and dark brown in color, giving the kitchen a rustic look. A microwave **is placed on top of a refrigerator**. There is **a refrigerator positioned in the middle of the kitchen, and a toaster and a bowl can be seen on a countertop**. The layout of the appliances creates an organized and functional kitchen space.

- - - - - - - - - - - - - - - - - - - - - - - - - - - - - - - - - - - - - - - - - - - - - - - - - - - - - - - - - - - -

**Rejected Response (Curriculum Strategy):**
The image showcases a small kitchen with various appliances and furnishings. There is **a small microwave situated in the middle of the kitchen** and **a sink can be seen in the right side of the kitchen. Numerous cabinets and cupboards** are present in the kitchen, providing ample storage space. **There is a refrigerator positioned in the middle of the kitchen, and a toaster and a bowl can be seen on a countertop**. The layout of the appliances creates an organized and functional kitchen space.

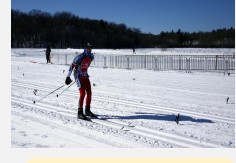

**Describe elements in the image in depth.**

**Raw Response:**
In the image, **a person is skiing down a snowy trail with several other skiers and snowboarders around.** The person in focus is **wearing a red and white outfit**, with **a pair of skis and poles visible in the scene**. The skier appears to **be skiing downhill in a fast and controlled manner. Other people** can be seen around the skiing area, **some of them also snowboarding or skiing**. The location seems to be a popular spot for winter sports, offering an exciting outdoor experience for the people involved.

**Preferred Response:**
In the image, **a person is skiing down a snowy path wearing a red and white outfit.** He **is holding ski poles and looking at the camera.** The scene also includes **several other people nearby**, creating a lively atmosphere, further emphasizing the popularity of winter sports in this location.

- - - - - - - - - - - - - - - - - - - - - - - - - - - - - - - - - - - - - - - - - - - - - - - - - - - - - - - - - - - -

**Rejected Response (Greedy Strategy):**
In the image, a person is skiing down a snowy trail, **wearing a red and white suit** and carrying skis and poles. **The skier appears to be skiing downhill in a fast and controlled manner. Several other skiers and snowboarders** are also present in the scene, **some of them riding their snowboards down the hill.** The location seems to be a popular spot for winter sports, providing an exciting outdoor experience for the people involved.

- - - - - - - - - - - - - - - - - - - - - - - - - - - - - - - - - - - - - - - - - - - - - - - - - - - - - - - - - - - -

**Rejected Response (Curriculum Strategy):**
In the image, **a person is skiing down a snowy path wearing a red and white outfit.** He **is holding ski poles and looking at the camera. Several other skiers and snowboarders** are also present in the scene, creating a lively atmosphere, further emphasizing the popularity of winter sports in this location.

Figure 10: **Qualitative Results of Preferred and Rejected Responses Generated by TPR and TPR-CL. Correct answers** and **hallucinations** are highlighted in color respectively.

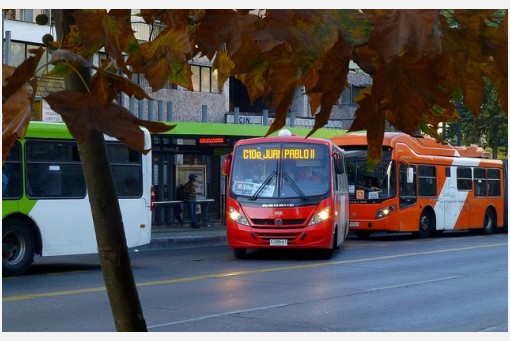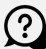

Detail the image's components with particular focus on each entity.

**TPR-7B (Ours):**

The image features a city street with **multiple buses parked or driving along the road**. The buses are of various colors, **including red, orange, green, and white.** They are lined up next to each other, creating a sense of a busy urban transportation scene. **A person holding a handbag is standing by the bus, perhaps waiting for the bus. The street is lined with trees**, adding a touch of nature to the urban setting. **The street appears to be well-maintained**, and the buses seem to be an essential part of the city's public transportation system. **The presence of multiple buses on the street** suggests that it could be a popular route or a hub for public transportation in the city.

- - - - - - - - - - - - - - - - - - - - - - - - - - - - - - - - - - - - - - - - - - - - - - - - - - - - - - - - - - - - - -

**LLaVA-1.5-7B:**

The image depicts a busy city street with **two buses driving down the road**. **One of the buses is red and white, while the other is orange and white.** Both buses are traveling in the same direction, with the red and white bus slightly ahead of the orange and white bus. **There are several people visible on the street, with some standing near the buses and others walking along the sidewalk. One person is carrying a handbag, adding to the bustling atmosphere of the city.** The scene captures a typical day in an urban environment with public transportation and pedestrians going about their daily routines.

- - - - - - - - - - - - - - - - - - - - - - - - - - - - - - - - - - - - - - - - - - - - - - - - - - - - - - - - - - - - - -

**LLaVA-NeXT-34B:**

The image depicts a city street scene with a focus on public transportation. There are three buses visible: **1. In the foreground, there is a red bus with a digital display on the front that reads "C10 JUR PABLO II." This suggests that the bus is on the C10 route and is heading towards a stop named "PABLO II."** 2. **In the middle of the image, there is a white bus with a digital display** on the front that reads "C10 JUR PABLO II." This indicates that this bus is also on the C10 route and is heading towards the same stop as the red bus. **3. In the background, there is an orange bus with a digital display on the front that reads "C10 JUR PABLO II." This bus is also on the C10 route and is heading towards the same stop as the other two buses.** The buses are parked along the side of the street, which is lined with trees that have autumn-colored leaves, suggesting that the photo was taken during the fall season. There is a building with a green awning in the background, and **a person can be seen walking on the sidewalk near the building**. The presence of multiple buses on the same route indicates a well-developed public transit system, which is essential for reducing traffic congestion and providing accessible transportation options for the city's population.

Figure 11: **Qualitative Results of TPR Compared with LLaVA-1.5-7B and LLaVA-NeXT-34B.** **Correct answers** and **hallucinations** are highlighted in color respectively.

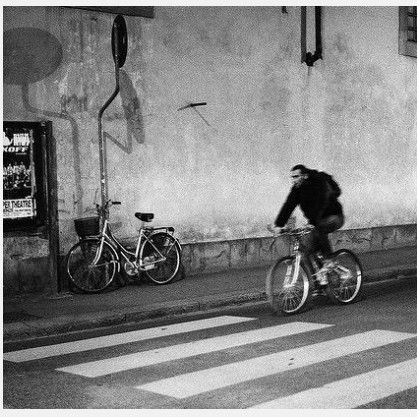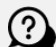

What are the main elements in this image? Describe them thoroughly.

**TPR-7B (Ours):**

The image depicts a man riding a bicycle on a city street at night. **The cyclist is riding down the street, possibly near a crosswalk**. There is **another bicycle parked or leaning against a wall** nearby, closer to the left side of the image. **The scene appears to be in black and white**, adding a sense of nostalgia or timelessness to the image. **The scene takes place at night**, which adds a sense of atmosphere to the image. The man riding the bicycle is the main focus of the scene, and the overall mood of the image conveys a sense of nighttime urban activity.

- - - - - - - - - - - - - - - - - - - - - - - - - - - - - - - - - - - - - - - - - - - - - - - - - - - - - - - - - - - - - - - - - - - -

**LLaVA-1.5-7B:**

The image depicts **a man riding a bicycle down a city street at night**. He is wearing a black jacket and appears to be the main focus of the scene. Another bicycle can be seen parked on the side of the street, closer to the left side of the image. **The street is lined with buildings on both sides, creating an urban atmosphere. There are also two traffic lights visible in the scene, one on the left side and the other on the right side of the street. Additionally, a handbag can be spotted near the center of the image, possibly belonging to the man riding the bicycle or someone else in the vicinity.**

- - - - - - - - - - - - - - - - - - - - - - - - - - - - - - - - - - - - - - - - - - - - - - - - - - - - - - - - - - - - - - - - - - - -

**LLaVA-NeXT-34B:**

The image is a black and white photograph capturing a moment on a city street. The main elements in the image are: 1. **A person riding a bicycle: The cyclist is in motion, pedaling down the street**. **They are** wearing a dark-colored jacket and appear to be carrying a backpack. **2. Two bicycles parked on the sidewalk: One bicycle is closer to the foreground, while the other is a bit further back. Both bicycles are leaning against the wall of a building.** 3. A street sign: **There is a street sign** visible in the image, although the specific details of the sign are not clear due to the angle and distance. 4. A building facade: **The wall of a building forms the backdrop of the image**. The building has a rough texture and appears to be made of concrete or a similar material. **5. Shadows and lighting: The image is well-lit, with shadows cast by the bicycles and the cyclist, suggesting that the photo was taken during the day with natural light.** Overall, the image conveys a sense of everyday life in a city, with a focus on cycling as a mode of transportation.

Figure 12: **Qualitative Results of TPR Compared with LLaVA-1.5-7B and LLaVA-NeXT-34B. Correct answers** and **hallucinations** are highlighted in color respectively.

