# OpenReview forum: "Systematic Reward Gap Optimization for Mitigating VLM Hallucinations"
_NeurIPS.cc/2025/Conference — NeurIPS 2025 poster_

### Official Review · Reviewer_Etoa · 2025-06-04

**Clarity:** 3
**Significance:** 3
**Originality:** 3
**Rating:** 4
**Confidence:** 4

**Summary:**

This paper introduces a novel DPO-based framework to mitigate hallucinations in VLMs, referred to as Topic-level Preference Rewriting (TPR). The methodology aligns well with the stated motivations, and the experimental results demonstrate the effectiveness of the proposed approach. Overall, I find this to be a solid piece of work that meets the acceptance threshold.

However, I have some questions and suggestions that I hope the authors will take into consideration. I am open to raising my score if these concerns are addressed satisfactorily.

**Questions:**

1. In Figure 1(b), what metric is utilized to quantify "hallucination"?
2. Suppose a response $y$ consists of two topics, $y_1$ and $y_2$, generated in sequence. The conditional probabilities $p(y_2|x, I, y_1)$ and $p(y_1|x, I, y_2)$ are generally not equivalent. How do the authors control the topic sequence in the proposed method to ensure consistency and coherence?
3. Do "Intra-Topic Self-Resampling" and "In-Context Rewriting" utilize prompts that are different (denoted as $x_1$ and $x_2$, respectively) from the prompt $x$ used during response generation? If so, $p(y|x, I)$ would likely differ from $p(y|x_1, I)$ and $p(y|x_2, I)$. Have the authors considered the impact of such differences on the proposed method?

**Ethical Concerns:**

["NO or VERY MINOR ethics concerns only"]

**Limitations:**

The limitations and broader impacts are discussed within the paper.
(I think the complexity of the proposed algorithm is also a limitation and should be included.)

**Paper Formatting Concerns:**

Not at all.

**Quality:**

3

**Strengths And Weaknesses:**

**Strengths:**
1. The paper is well-organized and easy to follow, with good writing quality.
2. The experiments demonstrate the effectiveness of the proposed method.
3. The concept of the "reward gap" is a valuable contribution, as it is often overlooked by DPO-based methods. The derivative curriculum learning approach makes a lot of sense and is well-justified.
4. The ablation studies are thorough and provide sufficient insights.

**Weaknesses:**
1. The proposed method involves multiple steps. Introducing them together in the form of a pseudo code or flowchart would greatly enhance readability and help readers better understand the methodology.
2. Some relevant references and baselines are missing, such as HA-DPO [1], OPA-DPO [2], and mDPO [3]. These works are closely related and should be properly cited and compared.
3. The experiments are conducted using only one base model (LLaVA-Instruct-1.5-7B). Extending the evaluation to larger models (e.g., 13B) would further strengthen the claims of the proposed method. However, I understand that time constraints during the rebuttal period may limit this. I hope the authors can include more results if the paper is accepted.

**Some minor issues:**
1. The paper does not include any equations. At the very least, the training loss for DPO should be introduced, as not all readers may be familiar with it. Additionally, in lines 128–130, it should be noted that DPO does not only align $\pi_\theta$ with the BT reward function; the KL divergence constraint is crucial, and the practical training loss of DPO cannot be derived without this term.
2. Please pay attention to the capitalization of article titles in the references. Only the first letter of the first word in a title and special abbreviations should be capitalized.

[1] Zhao et al. Beyond hallucinations: Enhancing LVLMs through hallucination-aware direct preference optimization.

[2] Yang et al. Mitigating hallucinations in large vision-language models via DPO: On-policy data hold the key

[3] Wang et al. mDPO: Conditional preference optimization for multi-modal large language models.

---

> ### Author Rebuttal · Authors · 2025-07-29
>
> ***Q1: Pseudo code / flowchart of proposed TPR.***
>
> Thanks for your advice. We will add a pseudo code in revised manuscript. It will be like:
>
> **Algorithm 1** Topic-level Preference Rewriting.
>
> **Require**: Reference model $\pi_\text{ref}$, Labeler model $\pi_\text{label}$, Source data $\mathcal{D}_\text{src}$ (Image $I$, Prompt $x$), Chosen strategy $\omega$ (e.g., greedy, curriculum).
>
> **Ensure**: Preference data $\mathcal{D}_\text{pref}=\{(I,x,y_w,y_l)\}$.
>
> 1:&nbsp;&emsp;Initialize $\mathcal{D}_\text{pref}\leftarrow\varnothing$;
>
> 2:&nbsp;&emsp;**for** each ($I$, $x$) in $\mathcal{D}_\text{src}$ **do**
>
> 3:&nbsp;&emsp;&emsp;&emsp;Initialize initial responses $S_y\leftarrow\varnothing$, semantic units $S_u\leftarrow\varnothing$, topic clusters $S_C\leftarrow\varnothing$;
>
> 4:&nbsp;&emsp;&emsp;&emsp;**for** $i\leftarrow$ 1 to $M$ **do**
>
> 5:&nbsp;&emsp;&emsp;&emsp;&emsp;&emsp;$y_i\leftarrow$ Sample($\pi_\text{ref}$, $I$, $x$);
>
> 6:&nbsp;&emsp;&emsp;&emsp;&emsp;&emsp;Add $y_i$ to $S_y$;
>
> 7:&nbsp;&emsp;&emsp;&emsp;&emsp;&emsp;Add Decompose($\pi_\text{ref}$, $y_i$) to $S_u$;
>
> 8:&nbsp;&emsp;&emsp;&emsp;**end for**
>
> 9:&nbsp;&emsp;&emsp;&emsp;$S_C\leftarrow$ TopicCluster($\pi_\text{ref}$, $S_u$);
>
> 10:&nbsp;&emsp;&emsp;&nbsp;**for** each cluster $C$ in $S_C$ **do**
>
> 11:&nbsp;&emsp;&emsp;&nbsp;&emsp;&emsp;IntraTopicResample($\pi_\text{ref}$, $C$);
>
> 12:&nbsp;&emsp;&emsp;&nbsp;&emsp;&emsp;Rank($\pi_\text{label}$, $C$);
>
> 13:&nbsp;&emsp;&emsp;&nbsp;**end for**
>
> 14:&nbsp;&emsp;&emsp;&nbsp;Initialize response template $y_k \leftarrow$ Randomly select from $S_y$, replacements $S_w\leftarrow\varnothing,S_l\leftarrow\varnothing$;
>
> 15:&nbsp;&emsp;&emsp;&nbsp;**for** each unit $u \in C$ in $y_k$ **do**
>
> 16:&nbsp;&emsp;&emsp;&nbsp;&emsp;&emsp;($u_w$, $u_y$) $\leftarrow$ SelectAlternatives($\omega$, $C$);
>
> 17:&nbsp;&emsp;&emsp;&nbsp;&emsp;&emsp;Add $u_w$ to $S_w$, Add $u_l$ to $S_l$;
>
> 18:&nbsp;&emsp;&emsp;&nbsp;**end for**
>
> 19:&nbsp;&emsp;&emsp;&nbsp;$y_w\leftarrow$ InContextRewrite($\pi_\text{ref}$, $y_k$, $S_w$);
>
> 20:&nbsp;&emsp;&emsp;&nbsp;$y_l\leftarrow$ InContextRewrite($\pi_\text{ref}$, $y_k$, $S_l$);
>
> 21:&nbsp;&emsp;&emsp;&nbsp;Add ($I$, $x$, $y_w$, $y_l$) to $\mathcal{D}_\text{pref}$;
>
> 22:&nbsp;&nbsp;**end for**
>
> 23:&nbsp;&nbsp;**return** $\mathcal{D}_\text{pref}$
>
> ***Q2: More relevant references and baselines (e.g., HA-DPO, OPA-DPO, and mDPO).***
>
> We compare TPR with HA-DPO/OPA-DPO/mDPO below.
>
> |Model|ObjHal-CHs↓|ObjHal-CHi↓|MMHal-Score↑|MMHal-Hall↓|AMBER-acc↑|AMBER-F1↑|POPE-Adv↑|POPE-All↑|RefoMB-Trust↑|RefoMB-Win↑|LLaVA-Bench↑|MMStar↑|
> |:----|:----:|:----:|:----:|:----:|:----:|:----:|:----:|:----:|:----:|:----:|:----:|:----:|
> |HA-DPO-7B|39.9|19.9|1.98|60.4|75.2|79.9|82.5|86.9|39.9|17.2|67.2|32.9|
> |OPA-DPO-7B|13.0|4.25|2.83|45.8|81.3|85.6|83.7|86.1|39.4|18.2|62.2|32.2|
> |mDPO-7B|35.7|9.8|2.39|54.2|-|-|-|-|-|-|-|-|
> |TPR-CL-7B|**3.4**|**1.8**|**3.06**|**30.2**|**82.7**|**87.8**|**84.2**|**87.6**|**61.6**|**32.3**|**71.1**|**33.3**|
>
> We will add these results and more references in revised version.
>
> ***Q3: Experiments on larger base model (e.g., LLaVA-1.5-13B).***
>
> We provide a comparison between 7B and 13B model on 2k/4k/8k/16k/20k data. The results are presented below.
>
> |Method|ObjHal-CHs↓|ObjHal-CHi↓|AMBER-acc↑|AMBER-F1↑|
> |:----|:----:|:----:|:----:|:----:|
> |7B baseline|53.6|25.2|73.5|77.6|
> |TPR-7B-CL 2k|41.7|20.5|75.2|79.5|
> |TPR-7B-CL 4k|18.8|10.2|77.1|81.6|
> |TPR-7B-CL 8k|12.0|5.9|78.9|83.4|
> |TPR-7B-CL 16k|5.0|2.5|80.8|85.5|
> |TPR-7B-CL 20k|3.4|1.8|82.7|87.8|
> |||||
> |13B baseline|46.3|22.6|75.3|79.2|
> |TPR-13B-CL 2k|27.5|13.6|78.8|82.1|
> |TPR-13B-CL 4k|9.1|4.7|81.4|85.2|
> |TPR-13B-CL 8k|5.5|2.7|84.6|88.6|
> |TPR-13B-CL 16k|2.6|1.6|85.7|89.6|
> |TPR-13B-CL 20k|2.1|1.0|86.6|90.2|
>
> The results show 13B model demonstrates a substantial and consistent reduction in hallucinations across all data scales, confirming TPR is highly effective on larger base models. We will add this table to revised manuscript.
>
> ***Q4: Details and Equation of DPO training loss.***
>
> Thanks for your advice. We will add equation of DPO loss and modify details about DPO in lines 128-130 in revised manuscript.
>
> ***Q5: Capitalization of article titles in the references.***
>
> Thanks for your advice. We will standardize the format of article titles in references in revised manuscript.
>
> ***Q6: About hallucination metric in Figure 1 (b).***
>
> In Figure 1 (b), we use an average of CHi/CHs in ObjHal as the metric to quantify "hallucination", because these two metrics simultaneously consider both coarse-grained and fine-grained hallucinatory contents.
>
> ***Q7: Suppose a response y consists of two topics, $y_1$ and $y_2$, generated in sequence. The conditional probabilities $p(y_2|x,I,y_1)$ and $p(y_1|x,I,y_2)$ are generally not equivalent. How do the authors control the topic sequence in the proposed method to ensure consistency and coherence?***
>
> * To handle topic sequence and ensuring coherence/consistency, our method addresses it through a two-fold mechanism grounded in Selective Topic Rewriting pipeline. Specifically,
>
>     * Our process does not generate topics in a new, arbitrary order. Instead, it is anchored by a response template, which is one of the initial, complete responses sampled from the reference model. This template naturally possesses a fixed topic sequence, e.g., $y=(y_1, y_2)$. During selective replacement, we perform an in-place modification of this template. Suppose we select alternative $y_1'$ for topic $Y_1$ and $y_2'$ for topic $Y_2$, then the new response $y'=(y_1',y_2')$ strictly preserves the original sequence. Therefore, the conditional dependencies remain consistent with template, i.e., the probability is always based on $p(y_1|x,I)$ and $p(y_2|x,I,y_1)$.
>     * In additional to preserving topic order, we also ensure logical and linguistic coherence. After the replacement units are chosen, we employ the "In-Context Rewriting" step. Here, we instruct the reference model itself to seamlessly integrate the chosen semantic units into the template, with explicit instructions to preserve logical structure and linguistic style. This step ensures the resulting response is fluent and natural.
> * For easy understanding, we have provided a simplified case study. **Correct contents** and *hallucinations* are highlight in different format respectively.
>     * Chosen Template: In the image, there is **a fork and a knife on the table** (Topic-1), ready for use. *There are two people in the scene, with one person located on the left side and the other person on the right side of the table* (Topic 2).
>     * Candidates for Topic-1: There is **a fork and a knife on the table**. / There are *two forks and one knife on the table*. / ...
>
>         Candidates for Topic-2: *There are two people in the scene, with one person located on the left side and the other person on the right side of the table*. / **Two people are located on the right side of the table**. / ...
>
>         \* Note that other initial responses may include phrases like "the water bottle is on the right side." These topics will also be broken down and clustered. But since the chosen template does not include this topic, it will be excluded subsequently. And we omit it here for clarity.
>     * Based on candidates on topic-1 and topic-2, we obtain the preferred and inferior response through rewriting the template with reference model.
>
>         Preferred Response: In the image, there is **a fork and a knife on the table**, ready for use. **There are two people in the scene, with both located on the right side of the table**.
>
>         Inferior Response: In the image, *there are two forks and one knife on the table*, ready for use. *There are two people in the scene, with one person located on the left side and the other person on the right side of the table*.
>     Throughout the entire process, the topic sequence remains consistent due to the template, and logical/textual coherence is also ensured through in-context rewriting.
> We will add this clarification in revised manuscript.
>
> ***Q8: Do "Intra-Topic Self-Resampling" and "In-Context Rewriting" utilize prompts $x_1$/$x_2$ that are different from prompts $x$ used during response generation? If so, $p(y|x,I)$ would likely differ from $p(y|x_1,I)$ and $p(y|x_2,I)$. Have the authors considered the impact of such differences on the proposed method?***
>
> * The prompt $x$ used for response generation is different from that used for self-resampling $x_1$ and in-context rewriting $x_2$. We provide the prompts in Appendix E (Table 5 and 6 in supp).
> * Consequently, the conditional probabilities $p(y|x,I)$, $p(y|x_1,I)$ and $p(y|x_2,I)$ are not identical. However, we argue that the impact of these differences is acceptable.
>     * Crucially, all three probabilities are sampled by the same reference model. While the prompts guide the model towards different sub-tasks, the generated text is always sampled from the same underlying model's capabilities and biases. The entire pipeline is therefore designed to explore the intrinsic success and failure modes of reference model itself.
>     * Different from methods using external models like GPT-4V for rewriting, TPR only introduces a minimal and controlled distribution shift. As shown in Figure 4 (c), the hallucination patterns introduced by external rewriters can differ significantly from the model's own failure modes. Our in-house approach avoids this mismatch.
>
> In summary, the impact of using different prompts is minimal and well-managed because the process remains self-contained within the reference model's distribution. We will add the discussion about this point in revised manuscript.
>
> ***Q9: Limitations of complexity of proposed method.***
>
> We will add complexity of proposed method as one of the limitation revised manuscript. We are committed to improving the framework's efficiency in future by simplifying the pipeline, e.g., investigating the necessity of the clustering step, to further enhance its throughput and efficiency.

---

> > ### Comment · Reviewer_Etoa · 2025-08-06
> >
> > I appreciate the authors for their detailed explanations, which have successfully addressed most of my questions and concerns. I look forward to seeing the updated version of the paper. I will maintain my positive score and recommend accepting this paper.

---

> > > ### Author Response · Authors · 2025-08-07
> > >
> > > Thanks for your reply. We will add the discussion improve the final version.

---

### Official Review · Reviewer_Aje6 · 2025-06-27

**Clarity:** 2
**Significance:** 2
**Originality:** 3
**Rating:** 4
**Confidence:** 4

**Summary:**

This paper addresses the challenge of mitigating hallucinations in vision-language models (VLMs) through reward-based methods, such as Direct Preference Optimization (DPO). To address the difficulty in optimizing the reward gap during dataa curation, the authors propose **T**opic-level **P**reference **R**ewriting (**TPR**), which selectively replaces topic-level content using model's internally resampled candidates. Addtionally, a curriculum learning strategy is employed to further improve TPR's effectiveness by gradually increasing the learning difficulty of by contructing hallucinated responses with progressively higher scores. Extensive experiments demonstrate that TPR achives the state-of-the-art performance across five hallucination benchmarks and two general benchmarks.

**Questions:**

1. **Clarification on Candidate Sampling Strategy**.
Could the authors clarify how TPR samples multiple candidate responses prior to topic-level decomposition? Specifically, what sampling strategy is used, and what temperature setting is employed during this process?

2. **Missing Comparisons on Data Efficiency**.
The paper appears to omit comparisons in data efficiency between TPR and certain baseline methods listed in Table 1, such as FGAIF and HSA-DPO. Could the authors explain the rationale behind this omission?

3. **Motivation for Topic-Level Rewriting**.
If I understand correctly, TPR is fundamentally a topic-level rewriting approach, which can be interpreted as a fine-grained method for optimizing hallucination reduction. Given this design choice, could the authors clarify why the "topic" level was selected over other potential levels of granularity (e.g., sentence-level or entity-level)? While topic-level decomposition is indeed common in preference learning, the definition of "topic" in this paper remains vague. For instance, in Figure 2, the elements labeled as "topics", such as weather, tower time, and tower style, do not clearly align with the conventional understanding of a "topic." It would be helpful if the authors could provide a more precise definition or criteria for what constitutes a "topic" in their framework.

4. **Difficulty Score used in Curriculum Learning**.
What 's the difficulty score used in TPR-CL in practice? Is it the reward gap?

**Ethical Concerns:**

["NO or VERY MINOR ethics concerns only"]

**Final Justification:**

The rebuttal has addressed most of my concerns.

**Limitations:**

Yes.

**Paper Formatting Concerns:**

There is a typo issue in line 348: "the s through".

**Quality:**

2

**Strengths And Weaknesses:**

Strengths:

1. **New Prospective**.
The proposed method TPR enhances the hallucination reduction performance from the prospectives of topic-level and progessively increasing the difficulty of hallucinated responses, which is different from previously published methods.

2. **Good Performance**.
TPR achives the state-of-the-art performance of hallucination reduction across popular hallucination benchmarks and general benchmarks. Furthermore, the results of ablation studies highlight the effectiveness of proposed method.

Weaknesses:

1. **Limited Technical Novelty**.
The technical contributions of the paper appear limited. The optimization method (TPR), rewriting strategies, and curriculum learning are all based on existing techniques. Additionally, the authors do not provide sufficient convincing evidence demonstrating why TPR outperforms other fine-grained approaches. It would strengthen the paper to include detailed comparisons or ablation studies to show whether the observed gains stem from the topic-level rewriting itself or simply from using a stronger rewriter model.

2. **Lack of Computational Cost or Time Cost Analysis**. The paper does not include any comparison of computational costs between the proposed TPR method and the other baseline approaches.

3. **Vague Definition of "Topic**. Another major concern is the vague definition of the term "topic" as proposed in this paper. To my knowledge, many fine-grained approaches have already been introduced to mitigate hallucination issues in VLMs, focusing on elements such as attributes, relations, and objects. However, the paper does not clearly articulate how "topic" differs from these existing fine-grained strategies.

---

> ### Author Rebuttal · Authors · 2025-07-29
>
> ***Q1: Technique novelty. Why TPR outperforms other fine-grained methods? Ablations on performance gains, from topic-level rewriting or a stronger rewriter model?***
>
> * **Technique Novelty**. We disagree the technical contribution is limited.
>     * Topic-level selective rewriting to construct $y_w$ and $y_l$ with deliberate semantic differences is a new data curation paradigm. To our knowledge, **this is the first work to actively sculpt the desired reward gap within preference data during data curation, rather than passively ranking VLM outputs or relying on black-box rewriters.**
>     * While TPR integrates established techniques like DPO and curriculum learning, we argue that their novel application for the purpose of systematically optimizing the reward gap configuration constitutes a significant contribution to mitigating VLM hallucinations. It's analogous to how major advancements are often achieved by innovatively combining mature methods. E.g., recent breakthroughs in reasoning models apply reinforcement learning to stimulate chain-of-thought processes. The value of such work lies not in inventing RL or CoT, but in their novel and effective application to unlock new capabilities, a principle that applies directly to our work.
>     * Some work (e.g., β-DPO) in other domains highlight the importance of reward gap in DPO, but they primarily address it by tuning hyperparameters in loss function, leading to potential distribution shift and uncontrolled results (see response to Reviewer zQZ4 ***Q2*** for details). In comparison, TPR provides  a mechanism to control the reward gap during data curation, avoiding above issues.
> * **Ablations on Performance Gains**. In Appendix D and Table 4 (supp), we provide experiments of using reference model (i.e., LLaVA-1.5-7B) itself as its labeler, named TPR-SL (self-labeling). It shows TPR-SL achieves substantial improvements over baseline LLaVA-1.5-7B model and remains highly competitive with other SOTA methods using much stronger labelers/rewriter like GPT-4V or LLaVA-NeXT-34B, demonstrating the performance gains are fundamentally driven by the design of TPR framework, and not solely by leveraging a more powerful labeler model.
> * **Comparison with Other Fine-grained Approaches**. In addition to above ablation study on performance gains, another evidence that TPR outperforms other fine-grained approaches is data efficiency.
>     * Compared to methods that do not rely on proprietary models (e.g., RLAIF-V), TPR constructs higher-quality responses through selective replacement. In contrast, while RLAIF-V improves the ranking process by aggregating the scores of all decomposed sub-responses, it still ranks existing, often flawed model outputs. This leads directly to the superior data efficiency we demonstrate in Figure 1 (b).
>     * Compared to other methods that applies human annotators for fine-grained labeling, such as RLHF-V, although they can produce high-quality data, they are prohibitively expensive and not scalable. TPR provides an automated, scalable framework that surpasses the performance of human-annotated data at larger scales, as shown in our data efficiency analysis.
>
> In summary, we believe TPR's technical contributions are significant, offering a new and effective perspective on data curation for preference alignment. This has also been recognized by other reviewers as a "novel" approach with "valuable contributions."
>
> ***Q2: Comparison of computation costs between TPR and other baseline approaches.***
>
> We provide a detailed breakdown and comparison below.
> * **Computation Cost**. The total computational cost for TPR generating full 20k dataset, run on 8 A100 GPUs, is detailed in the table below.
>     |Stage|TPR wo/ vLLM|TPR w/ vLLM|
>     |:----|:----:|:----:|
>     |Response Generation|10.27h|1.36h|
>     |Decomposition|8.91h|3.57h|
>     |Wh-question Convertion|7.95h|3.17h|
>     |Self-resampling|7.48h|2.96h|
>     |Topic Cluster|10.84h|6.71h|
>     |Scoring & Ranking|23.43h|7.42h|
>     |In-context Rewriting|2.17h|0.85h|
>     |**Total**|**71.05h**|**26.04h**|
>
>     With vLLM acceleration, the cost is just 4.7 GPU-seconds per pair.
>
> * **Comparison with Other Baselines**. We compare TPR with methods that curating data using open-sourced models (e.g., RLAIF-V) or proprietary models (e.g., POVID).
>     * To ensure a fair comparison with RLAIF-V, we use official RLAIF-V code to measure its computation cost under the same conditions (without vllm acceleration, using 8 A100 GPUs). It takes approximately 66 hours to produce 22k preference data in RLAIF-V. While RLAIF-V has fewer steps, the most computationally expensive step is the scoring phase which uses a 34B labeler model. This is a common practice shared by methods using open-sourced models, including TPR and RLAIF-V. Additionally, iterative refinement in RLAIF-V introduces an additional cost: intermediate fine-tuning, i.e., retraining policy model after each data generation round to serve as new reference model for next round. The training process is generally more resource-intensive than data generation (inference) steps and cannot be accelerated by inference engines. In comparison, given the similar computational budget, TPR can consistently deliver superior performance across multiple benchmarks.
>     * TPR is more cost-effective than methods relying on proprietary APIs. As detailed in Appendix C (Fig7 in supp), our analysis shows TPR achieves a comparable level of hallucination reduction at roughly one-third of estimated API cost required by a GPT-4V based rewriting approach.
>
> We will add this discussion in revised manuscript.
>
> ***Q3: Definition of "topic". Clarification why "topic" level was selected over other potential levels of granularity (e.g., sentence-level or entity-level).***
>
> * In TPR framework, the terminology "topic" is defined as a cluster of fine-grained semantic units that are determined as the same subject, based on textual consistency and visual correlation (see Section 3.2 and Appendix A).
> * We choose "topic-level" over other granularities because it best describes the flexible, emergent semantic clusters in TPR.
>     * **Why Not "sentence-level"**. The granularity of "sentence-level" is too coarse. The decomposed units in TPR are often segments of sentences, not whole sentences.
>     * **Why Not "entity-level" or "attribute-level"**. These granularities are too rigid. The resulting clusters are broader than just single entities or attributes. They inclues entities, relations, related implications. Forcing our data into such a structure (only entity-level or attribute-level) would be limiting and would fail to capture the full spectrum of potential hallucinations.
>     * **Why Use "topic-level"**. The term "topic" effectively conveys the key insight of our method. We are identifying the different subjects being discussed and then performing resampling and rewriting within the context of that same subject. Using a term like "segment-level" would fail to capture the crucial clustering step that is central to our contribution.
>
> We will clarify this definition of "topic" in revised manuscript.
>
> ***Q4: Details about candidate sampling strategy.***
>
> We follow RLHF-V and RLAIF-V to sample initial candidate responses. Specifically, we instruct the reference model with the prompt provided in the Appendix E (Table 5 in supp). We set the parameters as: `temperature=0.7,top_p=0.95,do_sample=True`. Other parameters like `top_k` are left at their default values. Besides, as detailed in Appendix E, we generated 10 candidate responses for each image. After these responses are decomposed into semantic units, we resample one additional alternative per topic.
>
> ***Q5: Data efficiency of FGAIF/HSA-DPO in Figure 1 (b).***
>
> The omission of FGAIF/HSA-DPO in Figure 1 (b) is because our local paper version was not updated to the latest during our experiments, so we cannot ascertain their precise data usage at that time. Given that they release their data in updated version, we have now conducted this comparison and will update our paper accordingly.
> * Note that both FGAIF and HSA-DPO rely on GPT-4V to collect auxiliary datasets for essential intermediate training steps, which represent a significant cost. To provide a fair comparison, we argue it is crucial to consider the total data volume leveraged by each method, not just the size of the final preference dataset (Otherwise, TPR can also collect additional dataset to train a stronger labeler model).
>     * FGAIF collects an auxiliary 3.5k dataset to train a reward model for its RL training stage, on top of its 14k preference data. The total effective data volume is about 17.5k.
>     * HSA-DPO uses a detect-then-rewrite pipeline that first requires a 16k dataset to train a hallucination detection model. In addition to 8k preference data used in DPO training, the total effective data volume is about 24k.
>
> * The results are present below. Note that we use average of CHs and CHi in ObjHal as the hallucination rate in Figure 1 (b), because these two metrics simultaneously consider both coarse-grained and fine-grained hallucination contents.
>
>     |Method|Data Scale|Hall (lower is better)|
>     |:----|:----:|:----:|
>     |TPR|16k|3.7|
>     |TPR|20k|2.6|
>     |RLAIF-V|22k|6.4|
>     |FGAIF|17.5k|5.1|
>     |HSA-DPO|24k|4.3|
>
>     As this data shows, both FGAIF and HSA-DPO perform similarly to RLAIF-V. Crucially, in the 16k-22k data range, TPR continues to demonstrate markedly superior data efficiency, achieving a significantly lower hallucination rate with less total data.
>
> We will update Figure 1 (b) in the revised manuscript to include these new data points. We thank the reviewer for pointing out this omission.
>
> ***Q6: Difficulty Score used in Curriculum Learning.***
>
> The "difficulty" in TPR refers to reward gap, it represents the learning difficulty.
>
> ***Q7: Some Typos.***
>
> We will fix typos in revised version. Thanks for your advice.

---

> > ### Comment · Reviewer_Aje6 · 2025-08-05
> >
> > Thank you for the authors' efforts in providing clarification. The rebuttal has addressed most of my concerns. Hope add the discussion into the latest version. I have raised my originial score.

---

> > > ### Author Response · Authors · 2025-08-05
> > >
> > > Thanks for your reply. We will take your advice to improve the final version.

---

### Official Review · Reviewer_o8b2 · 2025-07-02

**Clarity:** 3
**Significance:** 3
**Originality:** 3
**Rating:** 5
**Confidence:** 3

**Summary:**

This paper introduces Topic-level Preference Rewriting (TPR), a novel framework designed to systematically optimize the reward gap configuration within preference data to mitigate hallucinations in Vision-Language Models (VLMs). The core idea is to achieve fine-grained control over semantic details in VLM responses by selectively replacing semantic topics with the model's own resampled candidates. This precise control enables advanced data curation strategies, such as progressively adjusting the difficulty of rejected responses, which helps in sculpting an effective reward gap configuration to guide the model against challenging hallucinations. Comprehensive experiments demonstrate that TPR achieves state-of-the-art performance on multiple hallucination benchmarks, outperforming previous methods and exhibiting superior data efficiency.

**Questions:**

See weakness

**Ethical Concerns:**

["NO or VERY MINOR ethics concerns only"]

**Final Justification:**

The rebuttal resolves my concern and I decide to raise my score to accept.

**Limitations:**

Yes

**Quality:**

3

**Strengths And Weaknesses:**

## Strengths

---

* **Novel and Systematic Reward Gap Optimization**: The paper introduces a novel paradigm, Topic-level Preference Ranking (TPR), which systematically optimizes true reward gaps within preference pairs for Direct Preference Optimization (DPO). This approach offers fine-grained, topic-level control over the semantic details in Vision-Language Model (VLM) responses, representing a significant advancement over existing ranking or rewriting methods that frequently yield suboptimal reward gap configurations. Such systematic control is critical for achieving robust VLM alignment, particularly in mitigating hallucinations.

* **Significant Performance Improvement**: TPR demonstrates state-of-the-art performance across multiple hallucination benchmarks. Notably, the TPR-CL variant achieves an approximate 93% reduction in hallucinations on ObjectHal-Bench and around 41% on MMHal-Bench, alongside substantial improvements on AMBER and RefoMB. These empirical results underscore the efficacy of the proposed design and methodology.

* **Effective Curriculum Learning Strategy**: The authors leverage TPR's fine-grained control to implement TPR-CL, a simple yet highly effective curriculum learning strategy. This strategy progressively adjusts the difficulty of hallucinations present in rejected responses ($y_l$), akin to hard negative mining. This mechanism compels the model to discern subtle inconsistencies and fine-grained details, thereby enhancing its robustness against challenging hallucinations and consistently yielding superior performance.

* **Superior Data Efficiency**: TPR exhibits high data efficiency, a direct consequence of its capacity to curate high-quality preference pairs. While human-annotated data, such as RLHF-V, demonstrates initial efficiency, its inherent cost limits scalability. TPR's automated process enables a rapid reduction in hallucination levels with increasing data scale, ultimately surpassing human-annotated data in both performance and cost-effectiveness. Furthermore, other automated methods, such as RLAIF-V, often incur significantly higher computational overhead.

---

## Weakness

---

A primary concern pertains to the selection of LLaVA-NeXT-34B as the labeler model ($\pi_{label}$) for intra-topic ranking, particularly given that the policy model ($\pi_{\theta}$) and reference model ($\pi_{ref}$) undergoing fine-tuning are LLaVA-1.5-7B. The utilization of a substantially larger and presumably more capable model (34B parameters versus 7B parameters) to ascertain factual accuracy and non-hallucinatory content for generating preference pairs raises a significant question regarding the degree to which this constitutes a form of **knowledge distillation** from a larger model to a smaller one.

The paper posits that "evaluating fine-grained units tends to yield more reliable assessments, allowing even moderately capable models (e.g., $\pi_{ref}$ itself) to serve effectively as $\pi_{label}$ in TPR." However, the presented research lacks a direct **ablation study** empirically demonstrating how the claims would hold if a smaller, equivalently capable model (e.g., $\pi_{ref}$ itself, i.e., LLaVA-1.5-7B) were consistently employed as $\pi_{label}$ instead of LLaVA-NeXT-34B. This absence of a direct comparative analysis impedes the ability to fully disentangle the inherent benefits of the TPR framework from the potential advantages conferred by leveraging a significantly larger and presumably less hallucinatory model for the labeling process. Consequently, the extent to which TPR's observed performance gains are attributable to its architectural design versus the superior capabilities of the $\pi_{label}$ remains underexplored.

---

> ### Author Rebuttal · Authors · 2025-07-29
>
> ***Q1: Comparison between "utilization of a substantially larger and presumably more capable model for non-hallucinatory content generation or factual ascertion" and "knowledge distillation from a larger model to a smaller one." Ablation study on using a smaller, equaivalently capable model as $\pi_\text{label}$ instead of LLaVA-NeXT-34B.***
>
> * **Ablation on Self-Labeling Provided in Appendix**. For reviewer's concern about the absence of an ablation study on using reference model itself as labeler model, in fact, we provide this experiment and analysis in the supplementary material. As detailed in Appendix D and Table 4 (supp), we evaluate a variant named TPR-SL (self-labeling). The results show that TPR-SL achieves substantial improvements over the baseline LLaVA-1.5-7B model and remains highly competitive with other SOTA methods that use much stronger labelers like GPT-4o or LLaVA-NeXT-34B. This directly supports our claim that the significant performance gains are fundamentally driven by the architectural design of the TPR framework, and not solely by leveraging a more powerful labeler model.
>
> * **Using Capable Labeler**. Our decision to use LLaVA-NeXT-34B in our main experiments is primarily to ensure a fair and direct comparison with contemporary methods. leveraging a more capable model or human annotators for supervision is a standard and established practice in the training-based vision hallucination migitation literature (RLHF/RLAIF). For example, rewriting-based methods like POVID using proprietary models like GPT-4V. RLHF-V relies on costly human annotator. And ranking-based baseline like RLAIF-V also employs LLaVA-NeXT-34B as its labeler. By using the same 34B labeler, we provide a fair comparison that isolates the benefits of our novel data curation strategy against existing work.
>
> * **Surpassing the Labeler Model after Alignment**. Furthermore, the outcome of our alignment process goes beyond what is typically expected from knowledge distillation. We provide a comparison below.
>
>     |Method|ObjHal-CHs↓|ObjHal-CHi↓|MMHal-Score↑|MMHal-Hall↓|AMBER-acc↑|AMBER-F1↑|RefoMB-Trust↑|RefoMB-Win↑|
>     |:----|:----:|:----:|:----:|:----:|:----:|:----:|:----:|:----:|
>     |LLaVA-NeXT-34B|12.6|6.4|**3.31**|34.4|81.4|85.4|44.4|**32.8**|
>     |TPR-CL-7B|**3.4**|**1.8**|3.06|**30.2**|**82.7**|**87.8**|**61.1**|32.3|
>
>     We can find that the student model (TPR-CL-7B) outperforms its teacher (LLaVA-NeXT-34B) on most of the hallucination benchmarks, demonstrating that TPR is not merely distilling knowledge.

---

> > ### Comment · Reviewer_o8b2 · 2025-08-05
> >
> > Thanks for pointing out the supplemental table. It resolves my concern and I've raised my score.

---

> > > ### Author Response · Authors · 2025-08-05
> > >
> > > Thanks for your reply. We will add the discussion improve the final version.

---

### Official Review · Reviewer_zQZ4 · 2025-07-02

**Clarity:** 3
**Significance:** 3
**Originality:** 3
**Rating:** 4
**Confidence:** 3

**Summary:**

This paper addresses the problem of hallucination in vision-language models (VLMs) trained via preference learning. It argues that the effectiveness of Direct Preference Optimization (DPO) depends on the reward gaps between preferred and rejected responses in curated preference data, but current ranking- or rewriting-based curation strategies do not allow systematic control of these gaps.
The authors propose Topic-level Preference Rewriting (TPR): a data curation framework that decomposes responses into semantic topics, generates topic-level alternatives via self-resampling, and then constructs preference pairs by selective replacement of topics to control semantic differences deliberately. They further introduce a curriculum learning variant (TPR-CL) that gradually increases negative example difficulty.
Extensive experiments show that TPR (especially TPR-CL) achieves state-of-the-art hallucination mitigation on multiple benchmarks (reducing hallucinations by up to 93% on ObjectHal-Bench), while maintaining or improving general VLM capabilities.

**Questions:**

Can you quantify the total compute cost (e.g., GPU hours or forward passes per preference pair) and discuss feasibility when scaling to millions of pairs or much larger models?
How well does TPR handle more complex multimodal reasoning hallucinations, e.g., involving multi-step reasoning, ambiguous contexts, or commonsense knowledge?

**Ethical Concerns:**

["NO or VERY MINOR ethics concerns only"]

**Final Justification:**

I have thoroughly reviewed the authors' response and appreciate their comprehensive answers, which address my concerns. While I remain curious about how topic-level decomposition benefits complex reasoning tasks, this is not a major issue, so I vote to accept this paper.

**Limitations:**

The authors discuss technical limitations well (e.g., focus on simpler hallucinations, compute cost). However, they should explicitly address potential negative societal impacts, such as misuse of generated “wrong” responses for disinformation or the environmental cost of large-scale data curation.

**Quality:**

3

**Strengths And Weaknesses:**

Strengths:
1. Topic-level decomposition and self-resampling for systematic reward gap control is an original and well-motivated idea.
2. Outperforms ranking-based and rewriting-based baselines across multiple benchmarks. Curriculum learning variant (TPR-CL) shows particularly impressive gains.
3. Includes detailed ablations, comparisons with human-annotated data, and discussion of data efficiency.

Weakness:
1. Experiments target typical hallucination benchmarks, with complex multimodal reasoning tasks left for future exploration. It remains uncertain whether topic-level decomposition will work on more complex tasks.
2. The notion of reward gap optimization is compelling but under-defined mathematically. There's no theoretical grounding for why these gaps generalize better.
3. While TPR improves data efficiency per sample, its complex multi-step generation and scoring pipeline may limit practical scalability for truly large-scale training regimes or for models serving many languages and domains.

---

> ### Author Rebuttal · Authors · 2025-07-29
>
> ***Q1: Topic-level decomposition on more complex tasks, e.g., multi-step reasoning or commonsense knowledge.***
>
> There is currently no visual hallucination benchmarks for complex reasoning, so we use more complex general VLM benchmarks (i.e., MMMU/SEED-Bench) for evaluation, which to some extent can reflect hallucination of TPR and other methods exposed in complex reasoning scenarios. The results are present below.
> |Model|MMMU-valid (acc)↑|SEED-Bench (acc)↑|
> |:----|:----:|:----:|
> |LLaVA-1.5-7B|26.1|56.3|
> |RLHF-V|28.7|58.1|
> |Silkie-10B|29.1|58.0|
> |POVID-7B|29.6|58.3|
> |AMP-MEG-7B|26.4|42.8|
> |RLAIF-V-7B|27.4|57.9|
> |TPR-CL-7B|**30.2**|**59.6**|
>
> We will add these results in revised manuscript.
>
> ***Q2: Theoretical grounding for TPR.***
>
> We provide a mathematical grounding:
>
> 1. **DPO Gradient**
>
>     The gradient of the DPO loss function, $\nabla\_\theta\mathcal{L}\_\mathrm{DPO}$, with respect to the policy model's parameter $\theta$ is given by:
>
>     $\nabla\_\theta\mathcal{L}\_\mathrm{DPO}=-\beta\cdot\sigma(-\beta M_{\pi_\theta})[\nabla_\theta\log\pi_\theta(y_w|x)-\nabla_\theta\log\pi_\theta(y_l|x)].$
>
>     where $\sigma$ is sigmoid function, $\beta$ is trade-off hyperparameter, $M_{\pi_\theta}$ is policy model's estimated reward gap, defined as:
>
>     $M_{\pi_\theta}=\left(\log\frac{\pi_\theta(y_w|x)}{\pi_\mathrm{ref}(y_w|x)}-\log\frac{\pi_\theta(y_l|x)}{\pi_\mathrm{ref}(y_l|x)}\right)$
>
>     The gradient consists of a direction vector and a magnitude scalar, $\sigma(-\beta M_{\pi_\theta})$. This scalar term, with a value between 0 and 1, directly controls the magnitude of the gradient update and is dependent on the model's current estimated reward gap, $M_{\pi_\theta}$.
>
> 2. **Curating Data with a Desired True Reward Gap $\Delta r^\*$ In TPR**
>
>     Let $r^*(y,x)$ be true reward function that perfectly captures human preferences. For any pair ($y_w,y_l$) we curate, we define the true reward gap: $\Delta r^\*=r^\*(y\_w,x)-r^\*(y\_l,x)$. $\Delta r^\*$ is a fixed property of curated preference data pair that represents its difficulty. A high $\Delta r^\*$ is an "easy" problem, while a low $\Delta r^\*$ is a "hard" problem. Note that true gap is distinct from the model's estimated gap $M_{\pi_\theta}$ mentioned above. Using data with different true reward gaps has a direct impact on $M_{\pi_\theta}$. Let's compare two training strategies, assuming the same initial model:
>
>     * **Constant $\Delta r^\*$**. If we train model on data with a constant true reward gap (e.g., only "easy," high-gap pairs), the model will quickly learn to make $M_{\pi_\theta}$ large for these types of problems. As $M_{\pi_\theta}$ grows, the gradient magnitude $\sigma(-\beta M_{\pi_\theta})$ shrinks towards zero, leading to the learning signal diminishes.
>
>     * **Gradually Lower True Gap**. After model mastering easy problems (warm-up stage), it is presented with "hard" pairs. A lower true reward gap means more difficulty in distinguishing between $y_w$ and $y_l$. It leads to $M_{\pi_\theta}$ being close to zero for challenging pairs. Accordingly, an $M_{\pi_\theta}\approx0$ places the learning process in the steepest region of the loss curve, maximizing the gradient magnitude. This forces the model to learn the fine-grained details it was previously ignoring.
>
> 3. **How About Changing $\beta$?**
>
>     The $\sigma(-\beta M_{\pi_\theta})$ term indicates that one can also influence gradient by changing $\beta$, e.g., dynamically adjusts $\beta$ based on $M_{\pi_\theta}$. However, if we want to achieve similar results that creates a strong learning signal for hard preference pairs by dynamically changing $\beta$, we would aggressively lower the value of $\beta$. Note that $\beta$ is originally designed to control the strength of KL penalty during DPO learning. A low $\beta$ could induce an unforeseen distribution shift. In contrast, TPR uses reference model for every creative step, ensuring the process remains grounded in the capabilities of the original reference model.
> We will add this discussion on revised manuscript.
>
> ***Q3: Total computation cost.***
>
> We provide a detailed breakdown and comparison below.
> * **Computation Cost**. The total computational cost for TPR generating full 20k dataset, run on 8 A100 GPUs, is detailed in the table below.
>     |Stage|TPR wo/ vLLM|TPR w/ vLLM|
>     |:----|:----:|:----:|
>     |Response Generation|10.27h|1.36h|
>     |Decomposition|8.91h|3.57h|
>     |Wh-question Convertion|7.95h|3.17h|
>     |Self-resampling|7.48h|2.96h|
>     |Topic Cluster|10.84h|6.71h|
>     |Scoring & Ranking|23.43h|7.42h|
>     |In-context Rewriting|2.17h|0.85h|
>     |**Total**|**71.05h**|**26.04h**|
>
>     With vLLM acceleration, the cost is just 4.7 GPU-seconds per pair.
>
> * **Comparison with Other Baselines**. We compare TPR with methods that curating data using open-sourced models (e.g., RLAIF-V) or proprietary models (e.g., POVID).
>     * To ensure a fair comparison with RLAIF-V, we use official RLAIF-V code to measure its computation cost under the same conditions (without vllm acceleration, using 8 A100 GPUs). It takes approximately 66 hours to produce 22k preference data in RLAIF-V. While RLAIF-V has fewer steps, the most computationally expensive step is the scoring phase which uses a 34B labeler model. This is a common practice shared by methods using open-sourced models, including TPR and RLAIF-V. Additionally, iterative refinement in RLAIF-V introduces an additional cost: intermediate fine-tuning, i.e., retraining policy model after each data generation round to serve as new reference model for next round. The training process is generally more resource-intensive than data generation (inference) steps and cannot be accelerated by inference engines. In comparison, given the similar computational budget, TPR can consistently deliver superior performance across multiple benchmarks.
>     * TPR is more cost-effective than methods relying on proprietary APIs. As detailed in Appendix C (Fig7 in supp), our analysis shows TPR achieves a comparable level of hallucination reduction at roughly one-third of estimated API cost required by a GPT-4V based rewriting approach.
>
> We will add this discussion in revised manuscript.
>
> ***Q4: Practical scalability for large-scale training regimes or for models serving many languages and domains. Feasibility when scaling to millions of pairs or much larger models.***
>
> While TPR involves a multi-step pipeline, it is practical for large-scale regimes because:
> * TPR pipeline can be decoupled with parallelizable workflow: generation (response generation/decomposition/resampling), processing (scoring/ranking/selective replacement/rewriting) and training (curriculum learning). It allows us to execute data generation and processing steps in parallel across various languages, domains. The outputs can then be aggregated for a unified training phase, making the framework scalable for large-scale regimes.
> * Additionally, TPR is designed for the post-training phase for alignment, not pre-training. This phase inherently operates on a much smaller data scale, typically in the order of hundreds of thousands to millions of preference pairs, rather than the trillions of tokens used for pre-training.
>
> In summary, given TPR's parallelizable workflow and its application in less data-intensive alignment phase, we believe in TPR's practicality and scalability. Also, as detailed in our response the your question (***Q3***) on computation cost, the per-pair generation cost is manageable and can be significantly accelerated with inference engines like vLLM, further strengthening the case for its feasibility when scaling to millions of pairs or larger models. We are also committed to improving the framework's efficiency. Future work will explore simplifying the pipeline, for instance by investigating the necessity of clustering step, to further enhance its throughput and efficiency.
>
> ***Q5: Potential negative societal impacts, such as misuse of generated "wrong" responses or environmental cost of large-scale data curation.***
>
> We agree a discussion of these aspects is essential for responsible research. In revised manuscript, we will add our response to these concerns in Appendix G. It will include:
>
> * **Misuse of Synthetic Data**. The synthetic "wrong" responses generated by TPR, which are designed to contain hallucinations, could be misused for disinformation if taken out of context. To mitigate this risk, we commit to the following actions for any public release of our generated preference data:
>     * The data will be released under a strict research-focused license, which prohibits commercial use and derivative works intended for malicious purposes.
>     * The released dataset will be accompanied by a detailed README file. It will prominently state that the negative examples are synthetically generated for the purpose of model alignment, and are not suitable for any other application. We will explicitly warn against their use in any public informational systems.
> * **Environmental Cost**: As discussed in our response to ***Q4***, TPR is applied during the post-training alignment phase. While not zero, the environmental impact is therefore substantially lower than that of training a VLM from scratch. Our work on improving data efficiency and future plans to simplify the pipeline leads to a smaller environmental footprint.

---

> > ### Comment · Reviewer_zQZ4 · 2025-08-05
> >
> > I have thoroughly reviewed the authors' response and appreciate their comprehensive answers, which address my concerns. While I remain curious about how topic-level decomposition benefits complex reasoning tasks, this is not a major issue, so I vote to accept this paper.

---

> > > ### Author Response · Authors · 2025-08-07
> > >
> > > Thanks for your reply. We will add the discussion improve the final version.

---

### Note · Authors · 2025-08-13

Dear Area Chairs and Reviewers,

We would like to express our gratitude for thorough reviews and constructive feedback. Our rebuttals have successfully addressed reviewers' concerns.

**Key Contributions of TPR**
* **Novel Data Curation Paradigm**: TPR is the first framework to actively sculpt the desired reward gap between preference pairs during data curation. This offers fine-grained control over semantic differences, moving beyond passively ranking model outputs or using black-box rewriters.
* **Effective Curriculum Learning**: We introduce TPR-CL, a curriculum learning strategy that leverages TPR's precise control. By progressively increasing difficulty of negative examples, TPR-CL forces the model to learn finer-grained distinctions, further enhancing its robustness.
* **State-of-the-Art Performance**: Our extensive experiments show that TPR sets a new state-of-the-art on multiple hallucination benchmarks, demonstrating superior performance and data efficiency.

**Summary of Discussions**
|Question|Raised By|Our Response|
|:----|:----|:----|
|Performance on complex reasoning tasks|zQZ4|We provide results on MMMU/SEED-Bench, showing TPR maintains a performance edge.|
|Theoretical grounding|zQZ4|We supply a mathematical justification, connecting TPR to the gradient dynamics of DPO loss function.|
|Computational cost and scalability|zQZ4, Aje6|We present a cost breakdown, demonstrating TPR's cost is comparable to baselines like RLAIF-V and highly efficient with optimizations (4.7 GPU-seconds/pair).|
|Broader societal impacts|zQZ4|We commit to adding a dedicated appendix section discussing societal impacts of TPR.|
|Ablations on using $\pi_{ref}$ as labeler|o8b2, Aje6|We highlight experiments from appendix (TPR-SL) and provide new analysis showing TPR's gains stem from its core framework.|
|Technical novelties|Aje6|We clarify TPR's novelty lies in its unique data curation paradigm that actively controls the reward gap.|
|Performance on larger base models|Etoa|We add new experiments on a 13B model, which shows stronger performance and confirms TPR's benefits scale effectively with model size.|
|Additional baselines|Etoa|We include comparisons against methods (HA-DPO, etc.), further cementing TPR's state-of-the-art performance.|

Once again, we thank you for your time and valuable feedback. We will incorporate new experiments, analyses, and clarifications into final manuscript to further strengthen the paper.

Yours sincerely,

The authors of Paper #306.

---

### Decision · Program_Chairs · 2025-09-17

**Decision:**

Accept (poster)

**Comment:**

This paper addresses the problem of hallucination in VLMs trained via preference learning. It argues that the effectiveness of DPO depends on the reward gaps between preferred and rejected responses in curated preference data, but current ranking- or rewriting-based curation strategies do not allow systematic control of these gaps. So, the authors propose Topic-level Preference Rewriting (TPR), which is a data curation framework that decomposes responses into semantic topics, generates topic-level alternatives via self-resampling, and then constructs preference pairs by selective replacement of topics to control semantic differences deliberately.

The final scores from the reviewers are Accept (x1) and Borderline Accept(x3). Overall speaking, the reviewers find the method to be well-motivated, technically novel, with sufficient evaluations and significant performance improvement, and superior data efficiency. Most of the negative concerns (e.g., vague definition of “topic”, lack of computational cost, comparisons on more models, etc.) from the reviewers are well addressed during the rebuttal. So, it deserves an Accept.